# African green monkeys avoid SIV disease progression by preventing intestinal dysfunction and maintaining mucosal barrier integrity

Kevin D. Raehtz[1], Fredrik Barrenäs[2,3], Cuiling Xu[1,4], Kathleen Busman-Sahay[5,6], Audrey Valentine[1], Lynn Law[7,8], Dongzhu Ma[9], Benjamin B. Policicchio[10], Viskam Wijewardana[4¤a], Egidio Brocca-Cofano[4¤b], Anita Trichel[11], Michael Gale, Jr.[7,8,12], Brandon F. Keele[13], Jacob D. Estes[5,6], Cristian Apetrei[1,10☯]*, Ivona Pandrea[4,10☯]*

1 Division of Infectious Diseases, Department of Medicine, University of Pittsburgh, Pittsburgh, Pennsylvania, United States of America, 2 Department of Microbiology, University of Washington, Seattle, Washington, United States of America, 3 Department of Cell and Molecular Biology, Uppsala University, Uppsala, Sweden, 4 Department of Pathology, University of Pittsburgh, Pittsburgh, Pennsylvania, United States of America, 5 Vaccine and Gene Therapy Institute, Oregon Health and Science University, Portland, Oregon, United States of America, 6 Oregon National Primate Research Center, Oregon Health and Science University, Portland, Oregon, United States of America, 7 Department of Immunology, University of Washington, Seattle, Washington, United States of America, 8 Center for Innate Immunity and Immune Diseases, University of Washington, Washington, United States of America, 9 Department of Orthopedic Surgery, University of Pittsburgh, Pittsburgh, Pennsylvania, United States of America, 10 Department of Infectious Diseases and Microbiology, Graduate School of Public Health, University of Pittsburgh, Pittsburgh, Pennsylvania, United States of America, 11 Division of Laboratory Animal Resources, School of Medicine, University of Pittsburgh, Pittsburgh, Pennsylvania, United States of America, 12 Washington National Primate Research Center, University of Washington, Seattle, Washington, United States of America, 13 AIDS and Cancer Virus Program, Frederick National Laboratory of Cancer Research, Frederick, Maryland, United States of America

☯ These authors contributed equally to this work.
¤a Current address: IAEA Laboratories Seibersdorf, F Austria
¤b Current address: BlueSphere Bio, UPMC, Pittsburgh, Pennsylvania, United States of America
* apetreic@pitt.edu (CA); pandrea@pitt.edu (IP)

**Data Availability Statement:** All relevant data are within the manuscript and its Supporting Information files

## Abstract

Unlike HIV infection, SIV infection is generally nonpathogenic in natural hosts, such as African green monkeys (AGMs), despite life-long high viral replication. Lack of disease progression was reportedly based on the ability of SIV-infected AGMs to prevent gut dysfunction, avoiding microbial translocation and the associated systemic immune activation and chronic inflammation. Yet, the maintenance of gut integrity has never been documented, and the mechanism(s) by which gut integrity is preserved are unknown. We sought to investigate the early events of SIV infection in AGMs, specifically examining the impact of SIVsab infection on the gut mucosa. Twenty-nine adult male AGMs were intrarectally infected with SIVsab92018 and serially sacrificed at well-defined stages of SIV infection, preramp-up (1–3 days post-infection (dpi)), ramp-up (4–6 dpi), peak viremia (9–12 dpi), and early chronic SIV infection (46–55 dpi), to assess the levels of immune activation, apoptosis, epithelial damage and microbial translocation in the GI tract and peripheral lymph nodes. Tissue viral loads, plasma cytokines and plasma markers of gut dysfunction were also measured

**Funding:** This work was funded by grants from the National Institutes of Health (NIH)/National Center for Research Resources/National Institute of Diabetes and Digestive and Kidney Diseases/ National Heart, Lung and Blood Institute/National Institute of Allergy and Infectious Diseases (NIAID): R01 RR025781 (CA/IP), R01DK113919 (IP/CA), R01DK119936 (CA), R01 HL117715 (IP), R01 AI119346 (CA), R01 HL123096 (IP); base grants to the Oregon National Primate Research Center (ONPRC): P51OD011092 (JDE) and Washington National Primate Research Center (WNPRC): P51OD010425 (MG); NIAID/OD Contract HHSN272201800008C (MG); the Reagent Resource Support Program for AIDS Vaccine Development, Quality Biological, Gaithersburg, Maryland (DAIDS Contract N01-A30018) (MG); the WNPRC SVEU contract N01-AI-60006 (MG); and NCI/NIH contract HHSN261200800001E (BFK). Significant parts of this study were supported by start-up funds from the School of Medicine of the University of Pittsburgh. KDR and BBP were supported in part by the NIAID Pitt AIDS Research Training grant (T32 AI065380). Funders had no role in study design, data collection and analysis, decision to publish, or preparation of the manuscript.

**Competing interests:** The authors have declared that no competing interests exist.

throughout the course of early infection. While a strong, but transient, interferon-based inflammatory response was observed, the levels of plasma markers linked to enteropathy did not increase. Accordingly, no significant increases in apoptosis of either mucosal entero- cytes or lymphocytes, and no damage to the mucosal epithelium were documented during early SIVsab infection of AGMs. These findings were supported by RNAseq of the gut tis- sue, which found no significant alterations in gene expression that would indicate microbial translocation. Thus, for the first time, we confirmed that gut epithelial integrity is preserved, with no evidence of microbial translocation, in AGMs throughout early SIVsab infection. This might protect AGMs from developing intestinal dysfunction and the subsequent chronic inflammation that drives both HIV disease progression and HIV-associated comorbidities.

## Author summary

African nonhuman primates that are natural hosts to SIVs can provide us with unique insight into the pathogenesis of HIV disease due to their remarkable ability to avoid pro- gression to AIDS, despite high levels of viral replication. A key question of SIV pathogene- sis in natural hosts is whether the lack of disease progression is due to an exquisite ability to repair lesions occurring during the acute infection or to completely maintain the integ- rity of the mucosal barrier throughout the SIV infection. In pathogenic HIV/SIV infec- tions of humans and macaques, the mucosal integrity is compromised during acute infection, leading to leakage of gut microbial byproducts and to the occurrence of chronic local and systemic inflammation, which plays a crucial role in driving progression to AIDS. Our study shows that the mucosal barrier integrity is never lost in African green monkeys, thereby avoiding the effects of chronic inflammation and disease progression.

## Introduction

The virulence of lentiviral infections of nonhuman primates (NHPs) can vary widely, ranging from nonpathogenic to highly pathogenic, depending on the NHP species [1–3]. For instance, SIV infection is pathogenic in Asian NHPs, such as rhesus (RMs), pigtailed (PTMs) and cyno- molgus macaques [1,2] and, in the absence of antiretroviral therapy (ART), progresses to AIDS; therefore, macaques have been extensively employed as models of HIV/AIDS in humans [4–7]. Conversely, SIV infections are nonpathogenic in African NHPs, such as Afri- can green monkeys (AGMs), sooty mangabeys (SMs) and mandrills (MNDs) [2,8,9]. In these species, disease progression is highly uncommon, only occurring in a handful of animals which had greatly outlived their normal life expectancy [10–13].

 The exact reasons for such different clinical outcomes of SIV infections are only partially understood. In fact, pathogenic and nonpathogenic HIV/SIV infections share key features, most notably the high levels of acute viral replication and the robust steady-state replication for the remaining lifespan of the host, which results in higher plasma viral loads (VLs) in some natural hosts than in the majority of untreated chronically HIV-infected individuals [3,14–16]. Furthermore, as the primary target cell of SIV in African NHPs is the CD4$^+$ T cell, African hosts undergo a severe CD4$^+$ T cell depletion in the gut of the same order of magnitude as that observed in the HIV infections and pathogenic SIV infections [17–24]. Meanwhile, multiple

studies failed to identify major differences between the pathogenic and nonpathogenic SIV infection with regard to the humoral and cellular immune responses [2,3,5,25–28].

These shared characteristics suggests that the lack of disease progression in natural hosts is not the result of an attenuated viral infection. Furthermore, the sporadic cases of AIDS documented in African NHPs [11–13] and the observation that direct virus transfer from SIV-infected SMs or AGMs to macaques results in progression to AIDS [29–31] demonstrate that control of disease progression is indeed independent of the virus and relies on host adaptations. These adaptations likely occurred during the long term SIV-African NHP host coevolution [32–36], that resulted in host coadaptation to counter the deleterious consequences of the SIV infection [37,38]. Virus-host coevolution is demonstrated by phenotypic features of natural hosts that likely contribute to prevention of progression to AIDS [33–35,38–43]; specifically, relatively low levels of CCR5$^+$ CD4$^+$ T cell targets at mucosal sites [21,32,44,45] and downregulation of CD4 expression by the helper T cells as they enter the memory pool [44,46]. Additionally, the usage of alternative coreceptors, such as CXCR6, might contribute to the preservation of central memory CD4$^+$ T cells in natural host species, including AGMs and sooty mangabeys [42,47].

In addition to these phenotypic adaptations, the vast majority of the studies performed over the last two decades collectively indicate that the main factor behind the lack of disease progression in the natural hosts of SIVs is their ability to actively control chronic immune activation and inflammation [2,21,31,41,48–50], which are the main drivers of disease progression and mortality in HIV-infected subjects [51–54]. Indeed, AGMs, SMs, and MNDs have the ability to resolve immune activation at the transition from acute-to-chronic SIV infection, and this is the best correlate of the lack of disease progression in these species so far [41,48,55–57].

However, the keystone feature of the nonprogressive SIV infection in African NHPs is the lack of mucosal dysfunction, which allows them to avert microbial translocation [3,23,51,58]. In pathogenic HIV/SIV infections, microbial translocation occurs as a result of acute viral replication and proinflammatory responses causing extensive damage to the intestinal mucosa [59]. These breaches of epithelial integrity allow microbes and microbial byproducts to move from the intestinal lumen into the surrounding tissues and general circulation [60,61], inducing: (i) recruitment of target cells to the site of viral replication; (ii) exhaustion of major immune cell populations; and (iii) establishment of a chronic inflammatory state which persists indefinitely [51,59,60,62,63]. Collectively, these effects fuel the destruction of the gut mucosa, triggering further inflammation and T cell immune activation, which cause even more damage, creating a self-sustained vicious cycle, which does not depend on viral replication, as the chronic inflammation and contingent immune activation persist even in patients on long-term ART [64,65]. The microbial translocation-induced chronic inflammation is now widely considered to be not only the driving force of the progression to AIDS, but also the cause of numerous AIDS-associated comorbidities, including cardiovascular disease [51,52,66–70]. Therefore, understanding the mechanisms leading to the control of systemic inflammation in African NHP hosts of SIV could lead to new strategies to prevent both HIV disease progression, and HIV-associated comorbidities.

African NHPs were reported to maintain mucosal integrity throughout the chronic infection, as suggested by both the lack of microbial translocation during either acute or chronic SIV infection [22,23,35,58,71,72] and by cross-sectional analyses of the gut epithelium of chronically SIV-infected SMs, which failed to identify any mucosal lesions [62]. Yet, it is unknown whether mucosal integrity observed in chronically SIV-infected African NHP hosts results from an exquisite ability to repair the injury inflicted to the gut during acute SIV infection, or whether natural hosts can simply prevent loss of gut integrity throughout infection.

We previously demonstrated the importance of a healthy mucosal barrier integrity in natural hosts of SIVs. Experimentally-induced colitis through administration of dextran sulfate sodium to chronically SIV-infected AGMs led to increased viral replication and altered key parameters highly predictive of HIV disease progression [73]. In contrast, attempts to directly control microbial translocation [67] and/or inflammation [74] in chronically SIV-infected PTMs without reducing the preexisting mucosal damage resulted in only transient positive effects. We therefore hypothesized that preventing gut dysfunction by maintaining mucosal barrier integrity during the early SIV infection may be an overlooked, yet essential, element that enables the natural hosts to avoid microbial translocation and disease progression.

To test this hypothesis, we conducted an extensive *in situ* analysis of gut integrity in AGMs serially sacrificed throughout the acute and postacute SIV infection; we assessed the gut mucosa for multiple markers for immune activation, inflammation, apoptosis, disruption of the epithelium and presence of bacterial proteins. When possible, the same parameters were measured in immune cell subsets by flow cytometry. We also monitored the systemic levels of mucosal immune activation and inflammation throughout the follow-up. We report that low levels of immune activation, inflammation, and apoptosis act in concert to preserve the mucosal barrier integrity during SIV infection of AGMs, thereby avoiding microbial translocation, and the systemic chronic immune activation and inflammation that drives HIV disease progression, all of which are readily apparent when compared to chronically SIV-infected RMs. Our results are supported by findings from RNA transcriptomics showing that AGMs exhibit only extremely limited alterations in genes associated with immune activation, inflammation and damage to the gut epithelium.

## Results

### Study design

To thoroughly test our hypothesis that natural hosts of SIVs have the ability to maintain a healthy mucosal barrier throughout the course of early SIV infection, twenty-nine adult male AGMs were intrarectally challenged with $10^7$ copies of SIVsab92018. The inoculum consisted of diluted plasma collected from an acutely infected AGM, which had been established to be effective in a preliminary study [22]. Four unchallenged adult male AGMs were included as a control group. Apart from these 4 animals, which were euthanized uninoculated, each AGM was euthanized at a set time point postinoculation, with the time points covering both acute and early chronic SIVsab infection. They were divided into the following groups based on their predicted viremic status at the time of sacrifice: (i) preinfection (baseline); (ii) preramp [1–3 days postinfection (dpi)]; (iii) ramp-up (4–6 dpi); (iv) peak (9–12 dpi); (v) set-point (46–55 dpi). AGM groups and the necropsy time points are shown in S1 Fig.

Blood and various tissues were collected from the AGMs both pre- and postinoculation. At the time of each necropsy, numerous compartments were sampled from each AGM. The collected tissues were snap frozen for DNA/RNA for qPCR, histologically preserved for IS/FISH or collected for lymphocyte separation for flow cytometry (blood, gut and LNs only).

As our main focus was the integrity of the mucosal barrier at sites distal to the site of inoculation, jejunum and colon were extensively used for these experiments, as well as axillary LNs, which were used as sentinel sites representative for the systemic effects of infection. These tissues were extensively sampled from the AGMs and then they were either snap frozen for DNA/RNA extraction or histologically preserved. Immune cells were also isolated from fresh tissues as previously described [22,75].

## High levels of SIVsab replication in the gut and lymph nodes (LNs) parallel VLs in blood during the very early stages of infection

As damage to the gut is triggered by inflammation driven by local viral replication, we sought to assess the timing and magnitude of the viral replication in the gut and peripheral LNs; we also surveyed the establishment of systemic viral replication through the plasma. First, we quantified the plasma VLs and then we extracted DNA and RNA from the whole snap frozen tissue samples and quantified total vDNA and vRNA in the transverse colon, jejunum and axillary LNs (Fig 1).

In the blood, we were able to detect SIVagm starting at 6 dpi at 3–4 log vRNA copies/mL plasma (3/3 animals). Plasma VLs peaked at 9–13 dpi, reaching as high as 5–7 log vRNA copies/mL plasma (7/7 animals). By the establishment of set-point viral replication at 46–55 dpi, VLs had stabilized at around 4–5 log vRNA copies/mL plasma (4/4 animals). Blood from these set-point AGMs were also sampled at 13 dpi, with similar VLs to the peak AGMs (Fig 1).

vRNA first became detectable in the colon between 4–6 dpi, at 1–2 log vRNA copies/$10^6$ cells (3/7 animals). In the jejunum, vRNA ranged between 0–3 log vRNA copies/$10^6$ cells (4/7 animals). Similarly, vDNA became detectable in the colon between 4–6 dpi, at 0–2 log vDNA copies/$10^6$ cells (3/7 animals). In the jejunum, vDNA ranged between 0–1 log vDNA/ copies$10^6$ cells (4/7 animals). The VLs then peaked between 9–12 dpi, at 4–5 log vRNA copies/$10^6$ cells for both jejunum and colon; at the same time points, vDNA levels ranged between 2–4 log vDNA copies/$10^6$ cells in the colon and 3–4 log vDNA copies/$10^6$ cells in the jejunum. At the transition to chronic infection (46–55 dpi), vRNA levels ranged between 3–4 log in both jejunum and colon, with vDNA showing only a slight decrease from peak levels, to 2–3 log vDNA copies/$10^6$ cells in the colon and 2–4 log vDNA copies/$10^6$ cells in the jejunum.

To understand acute systemic SIVsab spread and replication dynamics in AGMs, we measured vRNA and vDNA levels in the axillary LNs. Here, vDNA was detectable in LNs as early as 1–3 dpi in 2 animals at ~2 log vDNA copies/$10^6$ cells. Later, LN VLs followed the same general pattern as the gut, with the 4–6 dpi VLs varying from 0–2 log vRNA copies/$10^6$ cells for vRNA and 1–2 log vDNA copies/$10^6$ cells for vDNA. Axillary LN VLs then peaked at 9–12 dpi, at 3–5 log vRNA copies/$10^6$ cells for vRNA, and 2–4 log vDNA copies/$10^6$ cells for vDNA. The transition to chronic infection VLs were 3–4 log vRNA copies/$10^6$ cells for vRNA and 1–2 log vDNA copies/$10^6$ cells for vDNA.

## During acute SIV infection of AGMs, CD4$^+$ T target cells transiently decrease in circulation, while remaining relatively stable in other tissues

As previously reported, we found that circulating CD4$^+$ T cells counts were significantly decreased during the preramp-up period (*p = 0.0039*), the ramp-up period (*p = 0.0156*), and the peak of viral replication (*p = 0.0156*). By the onset of chronic infection, CD4$^+$ T cells were partially restored, albeit below the preinfection levels (Fig 2A). However, this trend was not reflected in other tissues. Thus, CD4$^+$ T cells remained stable in the LNs, with little to no variation at any point during infection. In the jejunum, CD4$^+$ T cells were not significantly decreased from baseline and remained mostly unchanged throughout the course of acute SIV infection. The only exception was an appreciable drop in CD4$^+$ T cells in the jejunum by the set-point of viral replication, though without reaching significance (Fig 2A).

We also examined the levels of CCR5$^+$ CD4$^+$ T cells in blood, jejunum and axillary LNs. Circulating CCR5$^+$ CD4$^+$ T cells displayed a similar pattern as the total CD4$^+$ T cell population, with significant decreases during the preramp-up (*p = 0.0156*) and peak of viral replication (*p = 0.0312*). However, the levels of CCR5$^+$CD4$^+$ T cells did not undergo any significant alterations in the axillary LNs or the jejunum (S2B Fig) at any point during the course of infection.

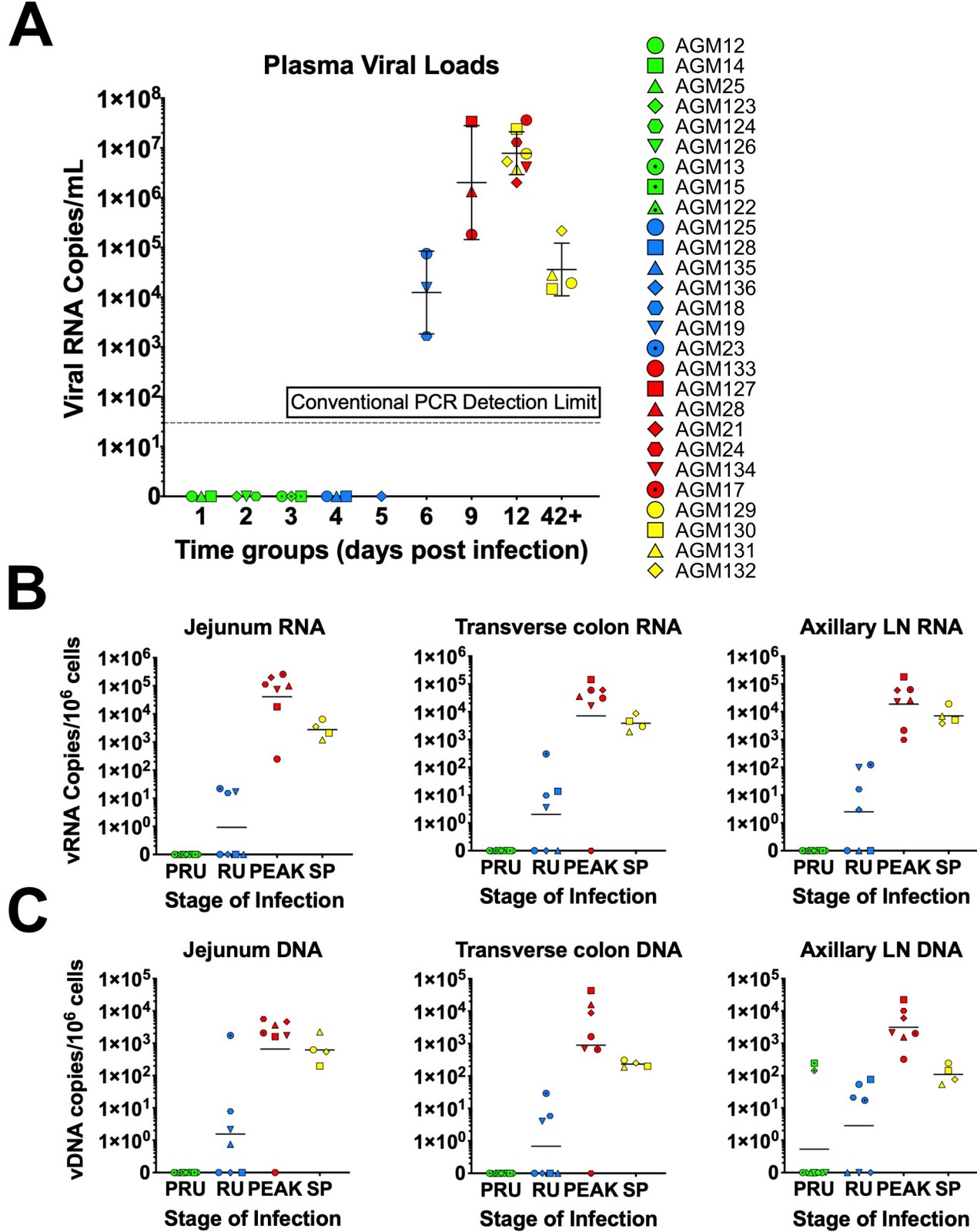

**Fig 1. Blood, jejunum and transverse colon tissue viral loads in SIVsab90218-infected AGMs.** AGM plasma viral loads (A) and the total number of copies of viral RNA (B) and viral DNA (C) per $10^6$ cells are shown for three tissues: jejunum, transverse colon, and axillary LN. Each individual animal is represented by symbol with a unique color and shape combination, as shown in the legend to the right of (A). The four groups are based on the days postinfection, with: PRU (preramp, 1–3 dpi), RU (ramp-up, 4–6 dpi), PEAK (peak, 9-12dpi) and SP (set-point, 46–55 dpi). Each group is

assigned a corresponding color: green (preramp), blue (ramp-up), red (peak) and yellow (set-point). The geometric mean value for each set of animals is also shown and the threshold for detection using qPCR is shown by a dashed line.

Similarly, CD8+ T cells and CD20+ B cells were transiently decreased in circulation but were largely unaltered in other tissues (S2A and S2B Fig). However, dendritic cells (DCs), monocytes and natural killer (NK) cells remain virtually unchanged during early SIV infection of AGMs (S3A and S3B Fig). The macrophage (Mφ) populations also showed no significant changes in the LNs or the gut (S3C Fig), yet, a significant, transient decrease in the absolute counts of circulating monocytes (Mo) occurred during ramp-up, with return to baseline or greater levels by set-point. Finally, the absolute NK cell counts decreased slightly, but significantly ($p = 0.0273$) in circulation during preramp-up, with no significant changes of the NK cell populations in the superficial LNs and gut NK cells (S3D Fig).

## Acutely SIVsab-infected AGMs exhibit minimal and transient T cell and B cell immune activation and proliferation

One of the key features of SIV infection in the natural hosts is the control of inflammation, immune activation and proliferation during the transition from acute-to-chronic SIV infection [2,3,28,48,56,57]. Using Ki-67 as a cellular proliferation marker, we monitored changes in

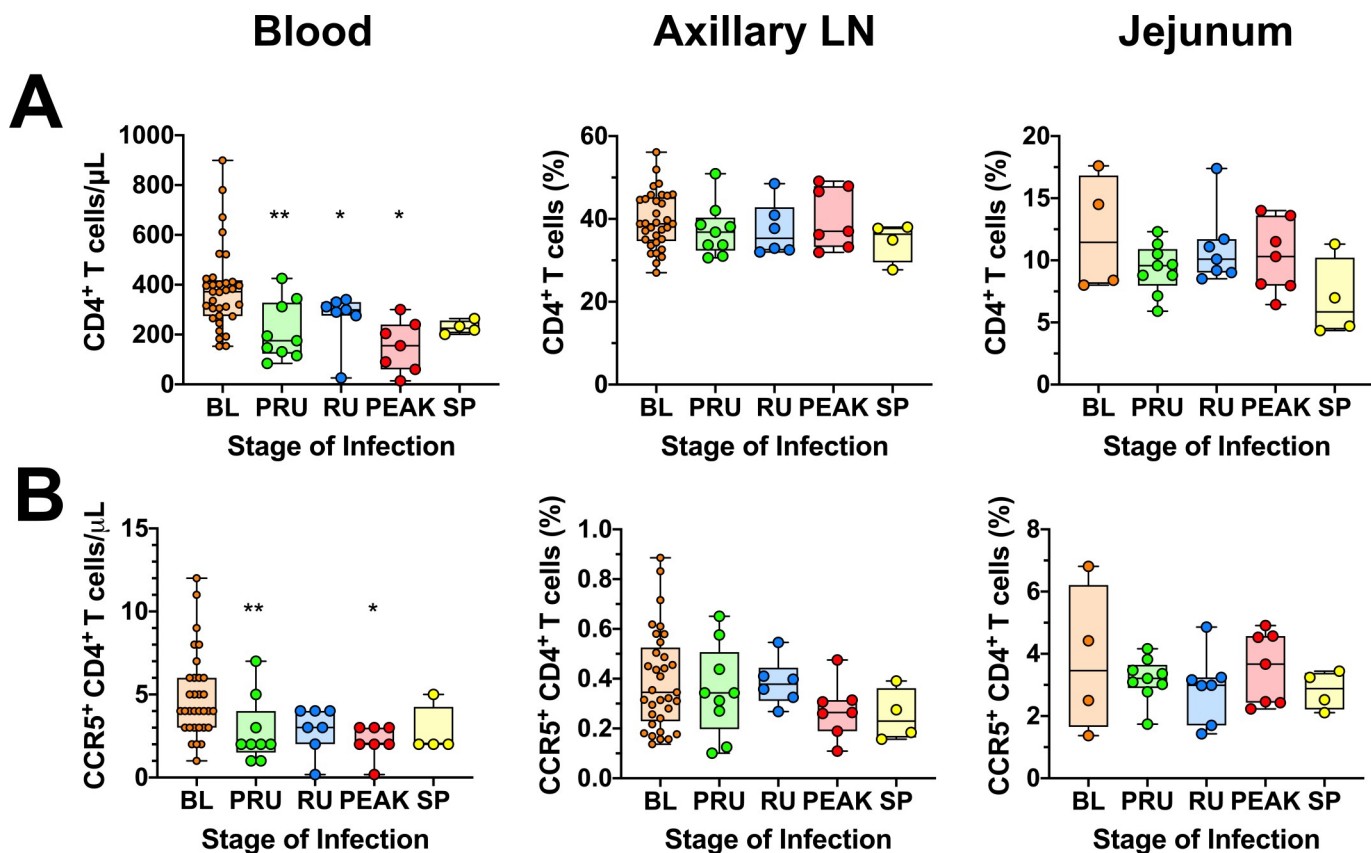

**Fig 2. CD4+ and CCR5-expressing CD4+ T-cell in the blood, jejunum and lymph nodes of SIVsab-infected AGMs.** Total populations of (A) CD4+ T cells; (B) CCR5+ CD4+ T cells isolated from blood, axillary LN and jejunum. The values for blood represent absolute cell counts, while the values in the jejunum and axillary LN represent percent populations. The five groups are based on the days postinfection, with: BL (baseline, preinfection, orange), PRU (preramp, 1–3 dpi, green) RU (ramp-up, 4–6 dpi, blue), PEAK (peak, 9-12dpi, red) and SP (set-point, 46–55 dpi, yellow). Asterisks indicates statistical significance when compared to baseline values, with * = p<0.05; ** = p<0.01.

these parameters for CD4$^+$ and CD8$^+$ T cells from blood, LNs, and gut via flow cytometry. The Ki-67$^+$ CD4$^+$ T cell fraction showed minimal changes at all these three sites (Fig 3A). There was a transient drop in the frequency of the Ki-67$^+$ CD8$^+$ T cells in the axillary LNs during the preramp-up ($p = 0.0078$) and ramp-up periods ($p = 0.0312$), with full restoration to baseline levels by the peak (Fig 3B). Ki-67$^+$ CD8$^+$ T cells then slightly increased at the set-point in the axillary LNs and, to a less extent, the jejunum, but did not reach significance.

As pathogenic SIV infections are associated with B-cell dysfunction [76], we next assessed the fate of CD20$^+$ B-cell populations during the earliest stages of SIVsab infection of AGMs,

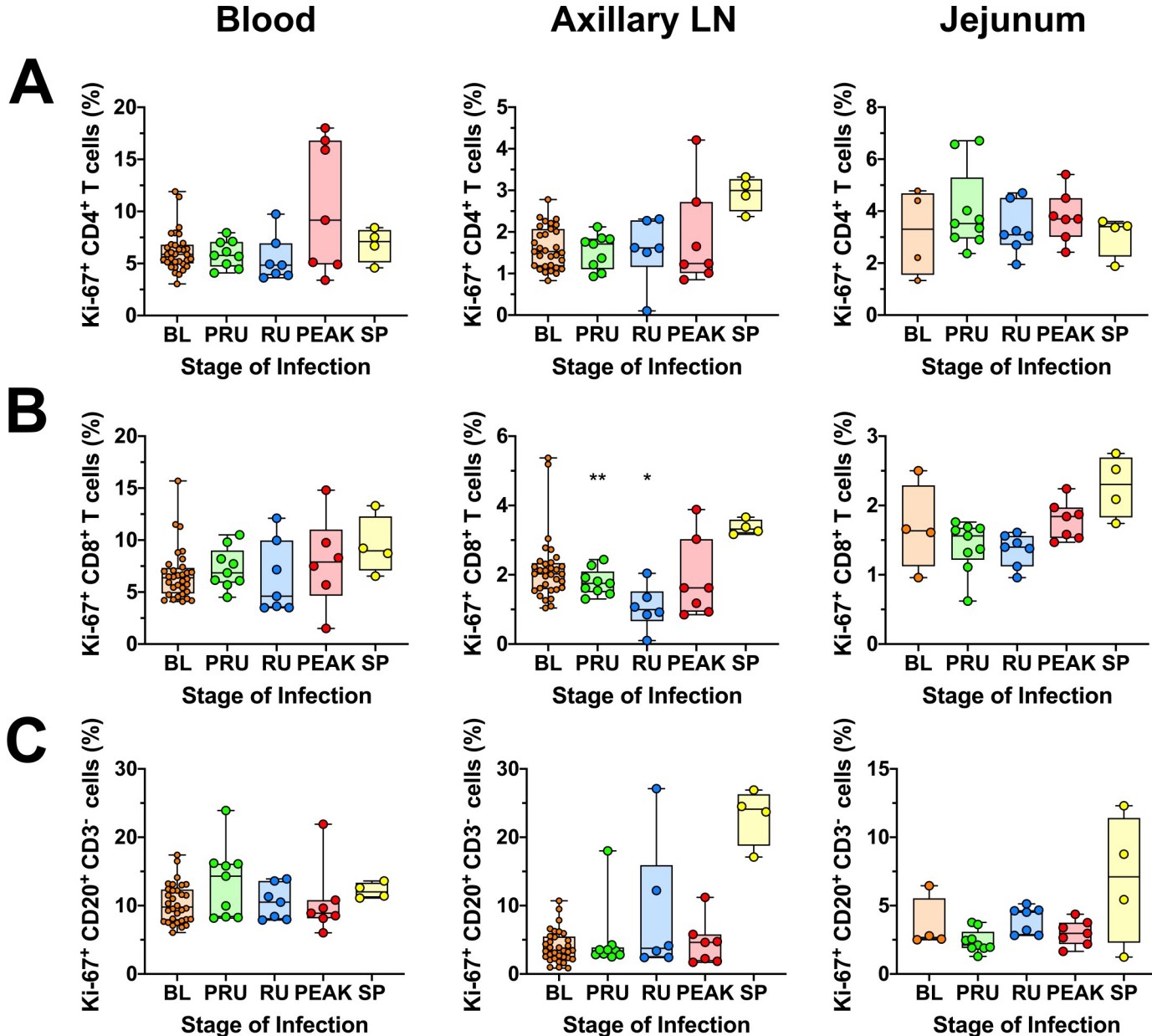

**Fig 3. General immune activation and proliferation in blood, jejunum and axillary lymph nodes in SIVsab-infected AGMs.** Ki-67 expression on (A) CD4$^+$ T cells; (B) CD8$^+$ T cells; and (C) CD20$^+$ B cells isolated from a variety of different tissues, including blood, jejunum and axillary LN. The five groups are based on the days postinfection, with: BL (baseline, preinfection, orange), PRU (preramp, 1–3 dpi, green) RU (ramp-up, 4–6 dpi, blue), PEAK (peak, 9-12dpi, red) and SP (set-point, 46–55 dpi, yellow). Asterisks indicates statistical significance when compared to baseline values, with * = $p<0.05$; ** = $p<0.01$.

and found that Ki-67$^+$ CD20$^+$ B cells remained relatively unchanged during acute infection in blood, LNs or gut, but increased in LNs, by early chronic infection, though this increase lacked significance. In the jejunum, there were no significant alterations of the CD20$^+$ B-cell activation or proliferation status, though some animals exhibited elevated levels by early chronic infection (Fig 3C).

## Transient increases of systemic inflammation occur during the acute SIVsab infection of AGMs

Plasma levels of a wide variety of cytokines and chemokines measured to assess systemic inflammation in acutely SIVsab-infected AGMs showed increases in two-waves, at 3–4 dpi and 9–12 dpi, with the second wave coinciding with the peak VLs. Several soluble markers of inflammation were consistently increased during the acute SIVsab and early chronic infection in AGMs [i.e., eotaxin (CCL11), IL-1RA (IL1RN), IL-8 (CXCL8), IP-10 (CXCL10), I-TAC (CXCL11), MCP-1 (CCL2), MIF (GIF), RANTES (CCL5)] (Fig 4). As our study design only encompassed the acute and postacute infection, before the inflammation markers were normalized, four chronically infected historic controls were included for this analysis. These AGMs were infected through the same ir route using the same SIVsab92018 stock at the same infectious dose, but were euthanized much later in infection, at 180 dpi [22]. In these controls, all plasma cytokines and chemokines were found to be at nearly baseline levels (Fig 4).

In addition to the systemic inflammation, we also surveyed markers of T-cell immune activation by measuring expression levels of CD69, HLA-DR and CD38 [75,77]. The levels of circulating and LN CD4$^+$ T cells expressing CD69$^+$ increased significantly, but transiently, either at preramp-up (axillary LN *p = 0.0391*) or peak (blood *p = 0.0156*), with CD69$^+$ expression back to baseline levels by viral set-point. Conversely, no significant increase in CD69$^+$ expression by CD4$^+$ T cells was observed in the jejunum (Fig 5A). Furthermore, the frequency of CD4$^+$ T cells expressing HLA-DR$^+$ and CD38$^+$ did not change significantly in any tissue at any time during acute or early chronic infection (Fig 5C), however, the HLA-DR$^+$ and CD38$^+$ expression by CD4$^+$ T cells was extremely variable across all groups. The levels of CD69$^+$ CD8$^+$ T cells showed no significant changes in the blood, axillary LN or jejunum at any point during the follow-up (Fig 5B). We did observe a significant, but transient, decrease in the HLA-DR$^+$ CD38$^+$ CD8$^+$ T cell levels in the blood during the pre-ramp (*p = 0.0195*) and ramp-up (*p = 0.0312*) and at the pre-ramp of infection in the axillary LNs (*p = 0.0039*), In all instances, CD8$^+$ T cell activation returned to baseline levels by viral set-point (Fig 5D).

## Levels of plasma markers of microbial translocation are not significantly altered by early chronic infection

Levels of various proteins in the plasma have been shown to be associated with microbial translocation during primary HIV/SIV infection, including: lipopolysaccharide (LPS) [68], soluble CD14 (sCD14) [78], and C-reactive protein [73]. To survey these markers in our AGMs we measured the plasma levels of these proteins during preinfection and the ramp-up, peak and set-points of viral replication using ELISA. We found no significant alterations during these time periods, apart from a transient decrease in sCD14 during the ramp-up (*p = 0.0312*) and peak (*p = 0.0312*) stages (S4A, S4B and S4C Fig). As microbial translocation is associated with chronic, systemic inflammation, we also measured the plasma levels of p-selectin, which is a marker of inflammation, endothelial activation and cardiovascular disease [79,80], and found no significant alterations (S4D Fig).

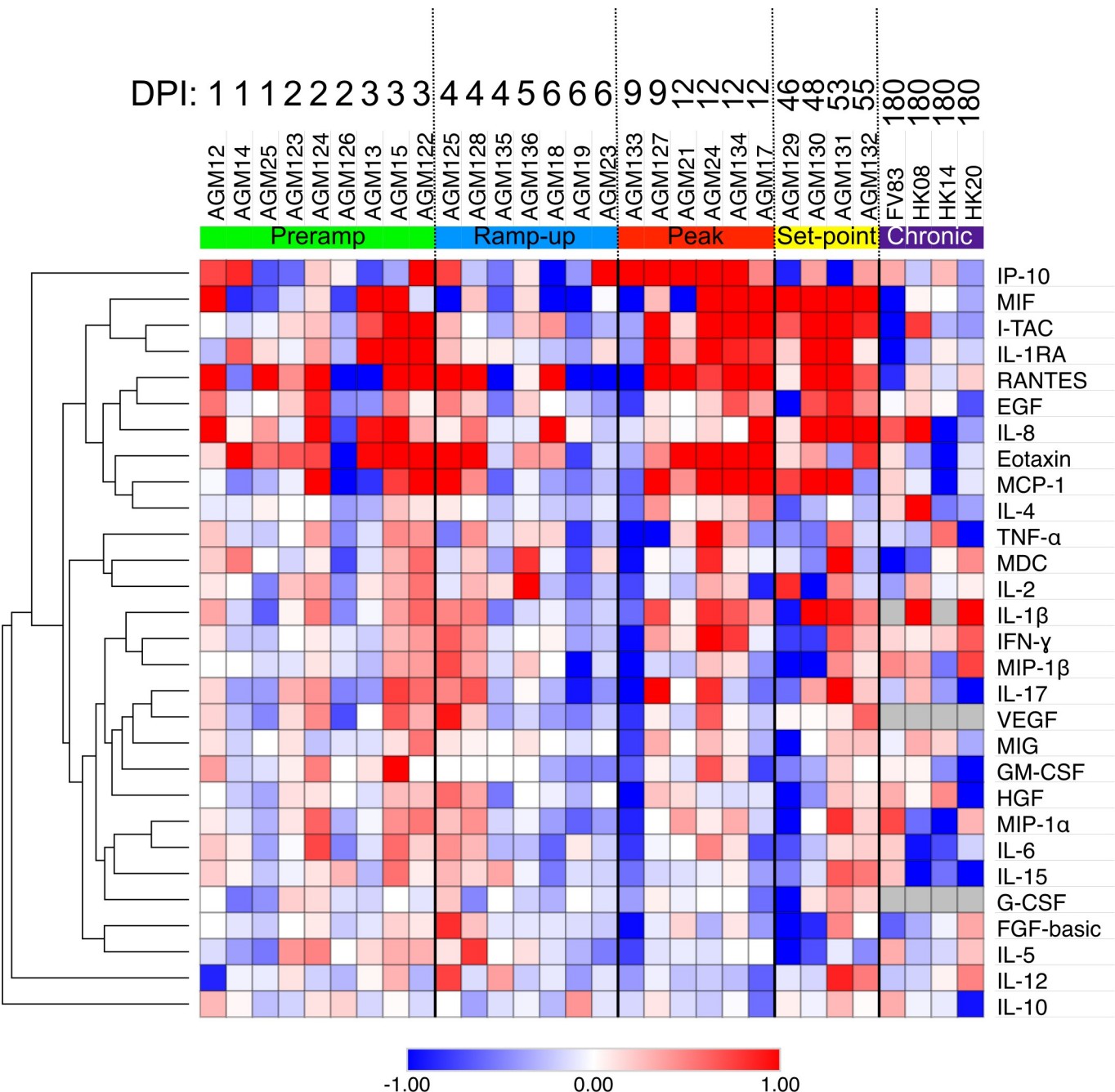

**Fig 4. Heatmap of cytokine fold changes from baseline following intrarectal SIVsab infection of AGMs.** Colors represent the fold change from preinfection to postinfection of the cytokines for each of the AGMs. Red indicates a positive fold change, while blue represents a negative fold change, with color intensity being proportional to the magnitude of the fold change, as shown below the heatmap. Animal numbers are shown above the heatmap along with dpi, with the colors below the numbers highlighting their time groups: orange (baseline), green (preramp), blue (ramp-up), red (peak), yellow (set-point), purple (AGM historical controls, late chronic). Cytokines and chemokines are listed on the right side of the heatmap. The results are clustered using a Spearman correlation, with the dendrogram showing relationship displayed on the left. The fold changes for the cytokines and chemokine levels were normalized with a $\log_2$ transformation. The heatmap was generated using the publicly available Morpheus software (Broad Institute).

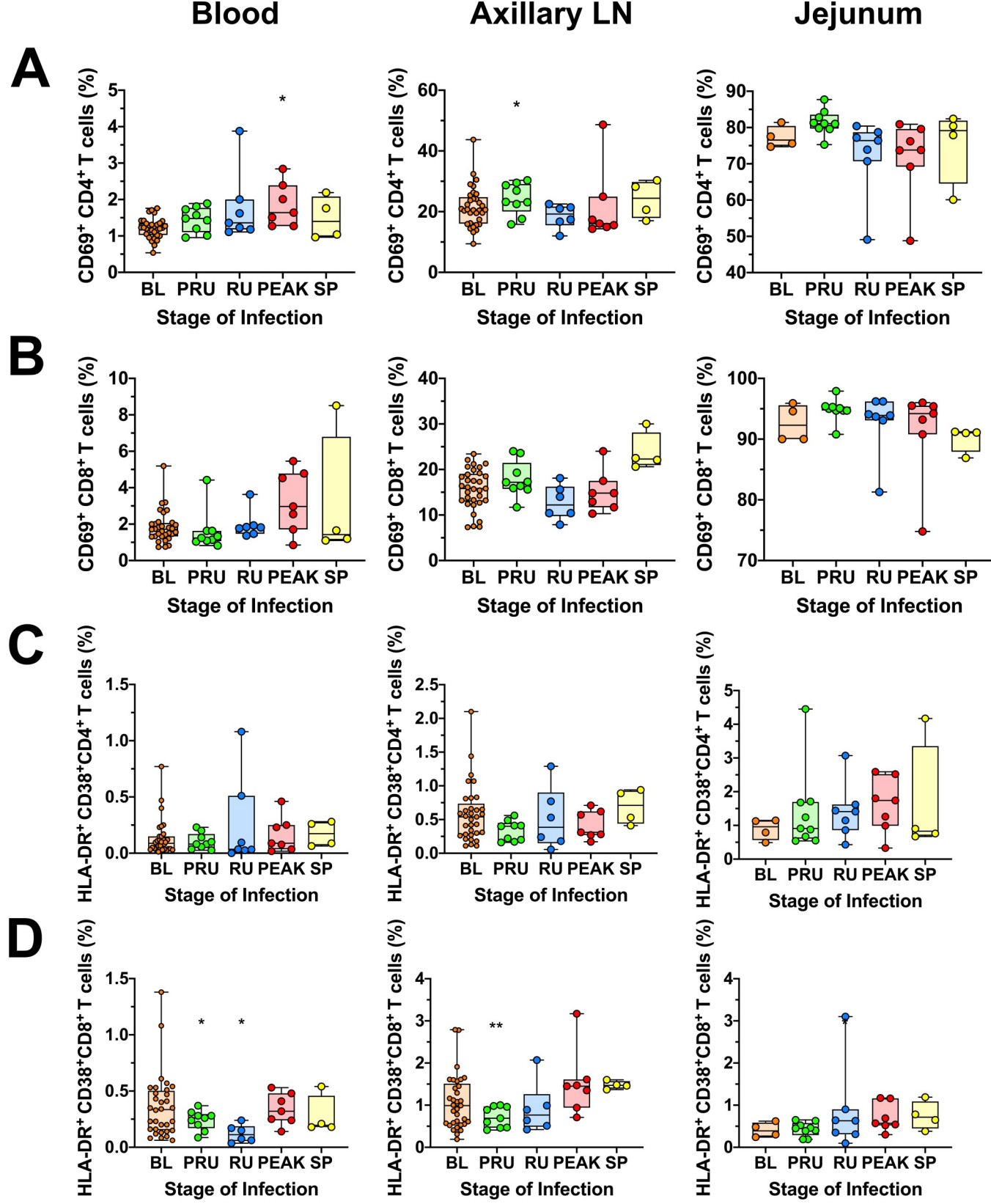

**Fig 5. Immune cell activation in blood, jejunum and lymph nodes in SIVsab-infected AGMs.** CD69 expression on (A) CD4+ T cells; (B) CD8+ T cells; and HLA-DR+ CD38+ expression on (C) CD4+ T cells and (D) CD8+ T cells isolated from a variety of different tissues, including blood, jejunum and axillary LN. The five groups are based on the days postinfection, with: BL (baseline, preinfection, orange), PRU (preramp, 1–3 dpi, green) RU (ramp-up, 4–6 dpi, blue), PEAK (peak, 9-12dpi, red) and SP (set-point, 46–55 dpi, yellow). Asterisks indicates statistical significance when compared to baseline values, with * = $p < 0.05$.

## Only transient mucosal inflammation and interferon-stimulated response occurs in acutely SIV-infected AGMs

In progressive HIV/SIV infection, mucosal inflammation and extensive cell death during the acute infection trigger multiple alterations to the gut integrity and microbial translocation, as well as systemic inflammation and immune activation, setting the stage for later disease progression [51,67,81,82]. Using IHC, we directly examined expression of three markers of mucosal immune activation/inflammation: (i) Ki-67; (ii) MPO, a major component of azurophilic granules, a defining attribute of neutrophils [83]; and (iii) MX1, an antiviral protein, directly linked to stimulation by type I and type II interferons [57,84,85]. There was no significant increase in the fraction of Ki-67 expressing cells in the lamina propria or the epithelium of the colon (Fig 6A and 6C) or the jejunum (S5 Fig). Conversely, in the LNs, Ki-67 expression was slightly increased at the peak and remained elevated into early chronic infection (Fig 6B). By comparison, during the chronic pathogenic infection of RMs, Ki-67 expression remained highly elevated in both the lamina propria and the mucosal epithelium (Fig 6A and 6C) of the gut, with large numbers of proliferating cells. This preponderance of Ki-67 was also reflected in the LNs, where there were far more Ki-67+ cells throughout the T cell zone, as demonstrated by quantitative analysis (Fig 6B and 6C).

There was a statistically significant ($p = 0.0153$), but transient increase in the frequency of MPO-positive neutrophils in the transverse colon during the ramp-up period (Fig 7A and 7C), that returned to baseline level by peak, with no significant changes in the number of MPO-expressing neutrophils in any of the other tissues studied (Fig 7A, 7B and 7C and S5 Fig). Note that AGMs have relatively high levels of MPO-positive cells in the colon prior to infection, higher than in uninfected RMs (which harbor virtually no MPO-positive cells in the lamina propria) [72], and in the range of some chronically SIV-infected RMs (Fig 7A), as supported by quantification (Fig 7C).

Conversely, MX1 expression increased sharply at the peak and rapidly returned to baseline level at the transition to chronic SIV infection in all examined tissues during the early SIVsab infection of AGMs (Fig 8A, 8B and 8C and S5 Fig). Notably, there was no significant difference between the MX1 levels at the peak of viral replication in AGMs and chronic infection in RMs, in either the gut or the LNs. By comparison, the MX1 expression in early chronic AGMs was significantly lower in the colon ($p = 0.0040$) and LNs ($p = 0.0015$) than in the chronically SIV-infected RMs (Fig 8A, 8B and 8C).

## AGMs show very limited changes in the expression of the genes associated with immune activation and epithelial damage in response to infection

A transcriptomic profiling of gut tissues taken during necropsy [86] was performed to assess changes in genes associated with epithelial damage, adaptive and innate immune responses and antiviral immunity (Fig 9A). Most of the surveyed genes showed only limited alterations. Notable exceptions include several CD4+ T cell-associated genes (including HPGDS and MYD88), which were transiently upregulated during the ramp-up and peak stages of infection. Two genes linked to CD8+ T cell regulation (including *HSDL1* and *FN1*) were also widely upregulated during the peak SIVsab infection.

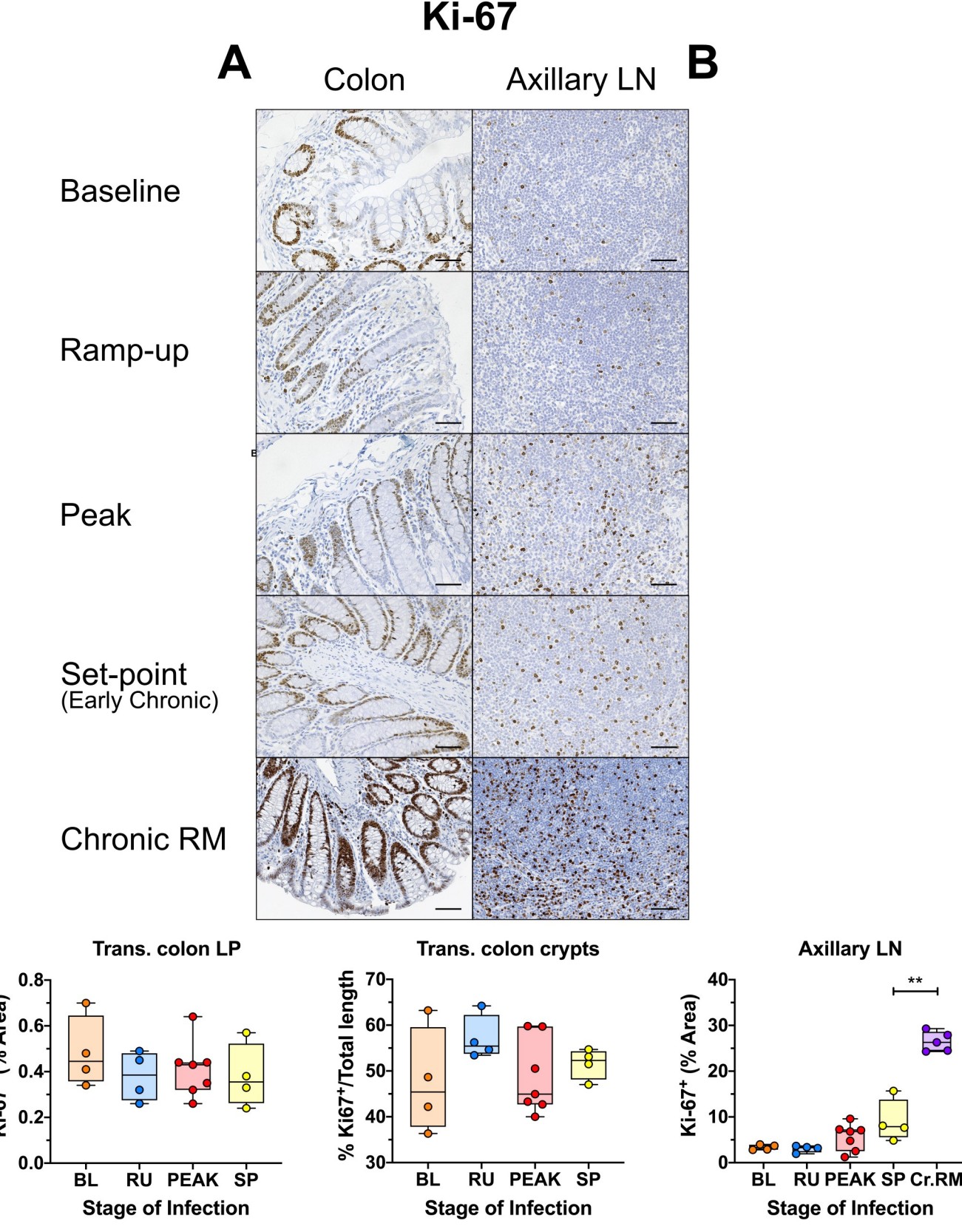

**Fig 6. Immunohistochemistry (IHC) for immune activation and proliferation in SIVsab-infected AGMs.** DAB-based IHC for Ki-67 in the (A) transverse colon, (B) axillary LN of AGMs and chronically SIV-infected RMs. In all the images, positive DAB signal is shown in brown, with the remaining tissue counterstained blue. The representative images of the gut display longitude cuts of the villi to better show the localization of proliferating epithelial cells in the crypts. Representative images of the axillary LN are taken from the T-cell area, to avoid the dense clustering of proliferating cells in the B cell follicles. Below are shown image quantifications (C) of proliferation in the gut and gut mucosa. For the colon, the ratio of the length of Ki-67 epithelial cells along the colonic crypts vs the total crypt length was used to estimate epithelial cell proliferation, while exclusion of the epithelium from the images was used to measure Ki-67 in the lamina propria alone. The quantification for each animal represents the average of the values from 9–12 individual image quantifications. The four different time groups are based on the days postinfection, with: BL (baseline, preinfection, orange), PRU (preamp, 1–3 dpi, green) RU (ramp-up, 4–6 dpi, blue), PEAK (peak, 9-12dpi, red) and SP (set-point, 46–55 dpi, yellow). The chronic RMs are shown in purple. All AGM quantifications were performed using FIJI version 1.0. Asterisks indicate statistical significance, with ** = $p < 0.01$. All AGM images were captured at 200X magnification using an AxioImager M1 bright-field microscope equipped with an AxioCam MRc5. z Scale bar: 100 μm.

Apart from T cell-associated genes, genes linked to CD20[+] B cell regulation were also analyzed, with no clear expression patterns being apparent and any variation appeared most likely to be due to interanimal variability (Fig 9A).

Several antiviral genes were also upregulated in response to infection (Fig 9A). Most notable amongst these was MX1 [57,84,85], but several other genes showed increased expression. Amongst these, CXCL12 showed consistent, low-level expression starting immediately following infection at 1 dpi. By blocking CXCR4, CXCL12 is known to inhibit viral entry and high plasma levels of CXCL12 are associated with nonprogression in HIV-infection, but may also stimulate proviral gene expression [87,88]. The levels of BST2 were also increased, mostly around the peak of infection. Also known as tetherin, BST2 can play an important role in HIV infection by preventing virus egress from cells. However, it is questionable whether it has a major impact on SIV-infection in AGMs [89,90].

We next established whether there were any interactions between these genes and their respective pathways. To this end, we linked established protein interactions between all included genes from the IPA database. The resulting network (Fig 9B) showed a central role for the transcription factor STAT3. However, most of the genes in the pathway, including *STAT3*, did not show any significant alteration in expression over the course of infection.

Finally, the gene expression analysis also showed that AGMs do have a strong antiviral response at the peak of infection. This is seen primarily through the highly increased expression of the interferon-induced genes *CXCL10* and *CXCL11* and *CXCL12*, which promote immune cell activation and migration, and MX1 (Fig 9 and S6 Fig). The increase in MX1 at the peak of infection mirrors the increases observed *in situ* in AGMs and RMs (Fig 8 and S5 Fig). Interestingly, the AGMs alone showed strong, sustained upregulation of *IL-17B*, a potentially anti-inflammatory cytokine, expression starting immediately postinfection (S6 Fig) [91].

## Little to no apoptosis occurs in the gut lamina propria or epithelium throughout the early stages of SIVsab infection of AGMs

We assessed the AGM capacity to maintain mucosal barrier integrity [3,4] by examining three different conditions associated with gut damage: apoptosis, fibrosis and loss of tight junctions between epithelial cells [92–95]. Apoptosis was measured in the gut and LNs by assessing the levels of active caspase-3, a central component of both the extrinsic and intrinsic apoptosis pathways [96]. Relatively few cells expressed active caspase-3 in all tissues surveyed, especially the axillary LNs (Fig 10A and 10B and S5 Fig). Using quantitative image analysis, we found a slight increase of the levels of active capase-3 in the transverse colon during the ramp-up, which did not reach significance and rapidly returned to baseline, similar to the transient neutrophil increase (Fig 10C). In jejunum and LNs, active caspase-3 levels remained largely unaltered (S5 Fig). In contrast, a large number of cells expressing high levels of active capase-3 were present in the gut and LNs of chronically SIV-infected RMs (Fig 10A and 10B).

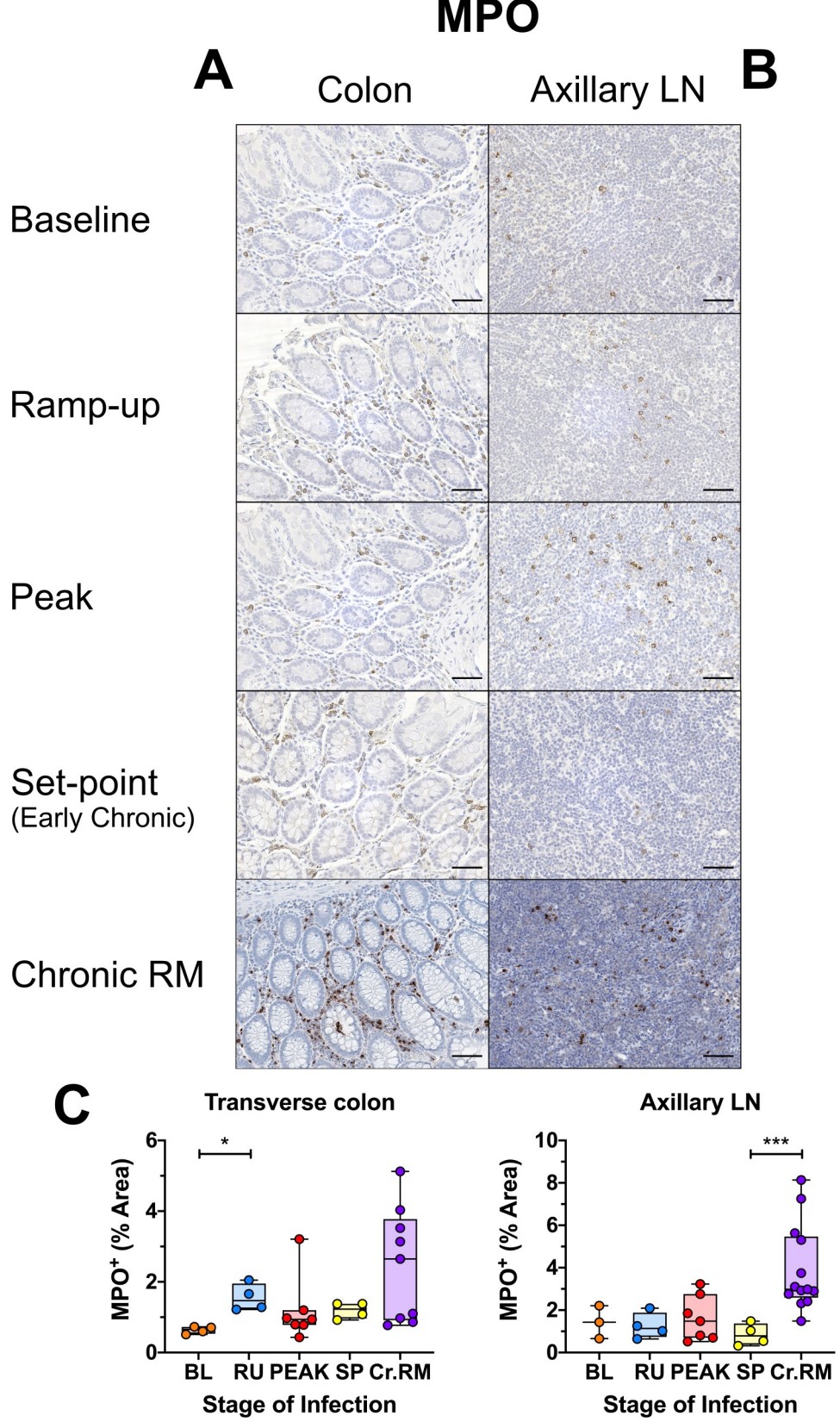

**Fig 7. Immunohistochemistry (IHC) for neutrophil infiltration and activation in SIVsab-infected AGMs.** DAB-based IHC for MPO in the (A) transverse colon, and (B) axillary LN of AGMs and chronically SIV-infected RMs. In all the images, positive DAB signal is shown in brown, with the remaining tissue counterstained blue. The representative images of the gut display lateral cuts of the crypts to demonstrate the presence of neutrophils in the surrounding lamina propria. Below are shown image quantifications (C) of the percent area of the total positive DAB signal. The quantification for each animal represents the average of the values from 9–12 individual image quantifications. The four different time groups are based on the days postinfection, with: BL (baseline, preinfection, orange), PRU (preramp, 1–3 dpi, green) RU (ramp-up, 4–6 dpi, blue), PEAK (peak, 9-12dpi, red) and SP (set-point, 46–55 dpi, yellow). The chronic RMs are shown in purple. All AGM quantifications were performed using FIJI version 1.0. Asterisks indicate statistical significance, with * = $p < 0.05$ and *** = $p < 0.001$. All AGM images were captured at 200X magnification using an AxioImager M1 bright-field microscope equipped with an AxioCam MRc5. Scale bar: 100 μm.

## Gut epithelial integrity is maintained throughout the course of SIV infection in AGMs

The preservation of epithelial integrity was assessed by staining for claudin-3, a principal component protein of cellular tight junctions, to visualize the continuity of the gut epithelium [97]. Importantly, no significant loss of continuity was observed in either the mucosal epithelium of the transverse colon (Fig 11A and 11B), or the jejunum (S5 Fig). The percent of damaged epithelium in the gut did not significantly change during SIV infection of AGMs, which is very different from the chronically SIV-infected RMs, where the percent of damaged epithelial barrier can be quite extensive and easily observable (Fig 11C and 11D), consistent with previous reports [62].

To further confirm that the gut epithelium did not sustain any significant injury during early SIVsab infection, we monitored the plasma levels of the enteropathy marker I-FABP. While the I-FABP levels (measured by ELISA) increased slightly in some animals during the preramp-up and ramp-up periods, this did not reach statistical significance and it quickly returned to baseline by the peak and remained at these levels during the early chronic infection (Fig 11E).

## Acute SIVsab infection does not induce fibrosis of the gut or LNs of AGMs

A hallmark of pathogenic HIV/SIV infections is the fibrotic damage of lymphoid tissues [98]. Therefore, we assessed whether SIV infection of AGMs is also associated with early gut and LN fibrosis, similar to pathogenic HIV/SIV infections [92]. Collagen levels in the colon, jejunum and peripheral LNs did not significantly change during SIV infection of AGMs (Fig 12 and S5 Fig), in contrast to the significant amount of collagen deposition in the LNs in RMs (Fig 12B and 12C). To further confirm the lack of fibrosis in SIVsab-infected AGMs, we monitored the plasma levels of hyaluronic acid, a biomarker of liver fibrosis [99]. Our results showed no perceptible changes in hyaluronic acid levels at any point of acute SIV infection, even at the peak of viral replication, with the exception of a few outliers (Fig 12D).

## No increased microbial translocation during early SIV infection in AGMs

To confirm the maintenance of mucosal integrity and prevention of microbial translocation in acutely SIV-infected AGMs, we performed quantitative IHC using (i) a monoclonal antibody to the core region of LPS, a highly immunogenic endotoxin found in the outer membrane of Gram-negative bacteria [100]; and (ii) a polyclonal anti-*Escherichia coli* antibody that cross-reacts with numerous enterobacteria species, previously used as markers of microbial translocation in pathogenic SIV infections [62]. Uninfected AGMs exhibited high levels of LPS in the transverse colon, largely localized to the lamina propria. These LPS levels were as high as those observed in some chronically SIV-infected RMs. However, the LPS did not increase

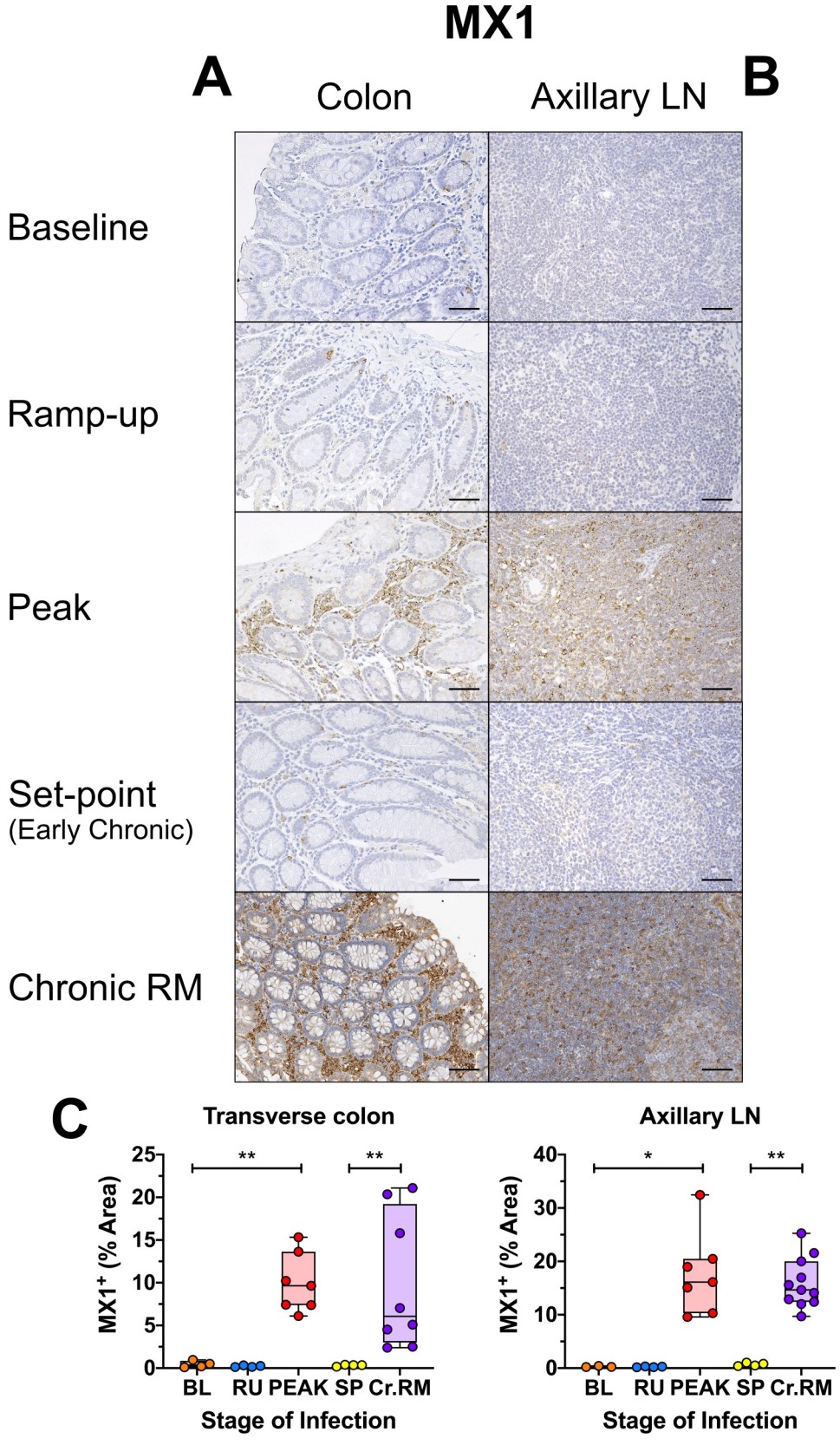

**Fig 8. Immunohistochemistry (IHC) to interferon-based responses to infection in SIVsab-infected AGMs.** DAB-based IHC for MX1 in the (A) transverse colon, and (B) axillary LN of AGMs and chronically SIV-infected RMs. In all the images, positive DAB signal is shown in brown, with the remaining tissue counterstained blue. Below are shown quantifications (C) of the percent area of the total positive DAB signal. The quantification for each animal represents the average of the values from 9–12 individual image quantifications. The crypts were included in the quantification and the crypt enterocytes may have contributed to the overall % area of positive. The four time groups are based on the days postinfection, with: BL (baseline, preinfection, orange), PRU (preramp, 1–3 dpi, green) RU (ramp-up, 4–6 dpi, blue), PEAK (peak, 9–12 dpi, red) and SP (set-point, 46–55 dpi, yellow). The chronic RMs are shown in purple. All AGM quantifications were performed using FIJI version 1.0. Asterisks indicate statistical significance, with * = $p < 0.05$ and ** = $p < 0.01$. All AGM images were captured at 200X magnification using an AxioImager M1 bright-field microscope equipped with an AxioCam MRc5. Scale bar: 100 μm.

postinfection in AGMs, indicating that the observed high levels are independent of SIV infection (Fig 13A). Image analyses and quantification of the LPS levels in the lamina propria at different stages of SIVsab infection in the colon revealed no significant change from the baseline (Fig 13C). Likewise, there were no significant changes in the amount of LPS in the jejunum or peripheral LNs either (Fig 13B & S5 Fig). In contrast, chronically SIV-infected RMs had abundant LPS-core present both beneath the epithelium in the gut (Fig 13A and 13C) and throughout the parenchyma of peripheral LNs (Fig 13B and 13C).

Using an additional stain to measure microbial translocation, a polyclonal anti-*Escherichia coli* antibody, we found that *E. coli* was present only very sparsely within the gut lamina propria (Fig 14A and S5 Fig), and at very low levels in the peripheral LNs throughout the course of infection in AGMs (Fig 14B). Quantitative image analysis demonstrated that none of the tissues displayed any significant changes in the *E. coli* levels during the course of SIV infection in AGMs. In contrast and, as with LPS, abundant levels of *E. coli* were present in both the gut lamina propria and peripheral LNs of the chronically SIV-infected RMs, with significantly higher levels of *E.coli* in the LNs than early chronic AGMs (Fig 14A, 14B and 14C).

## Discussion

African NHPs that are natural hosts of SIVs avert disease progression despite sustaining high levels of SIV replication by mechanisms that are only partially understood. It is widely acknowledged that understanding these mechanisms has the potential to generate new strategies for preventing HIV disease progression in humans [15,22,40,46,48,49,56,57,101–106], and could help alleviate the social and financial burden of HIV infection and long-term ART [26,107].

Here, we thoroughly assessed the impact of acute SIVsab infection in AGMs at the mucosal sites to understand if their ability to avoid disease progression stems from maintenance of a healthy gut throughout infection, or an exquisite capacity to rapidly resolve the effects of viral replication and T cell depletion at mucosal sites. In progressive lentiviral infections (i.e., SIV infection of macaques and HIV infection of humans), the majority of gut CD4+ T cells are initially killed by direct viral cytopathic effects due to ongoing viral replication, which drives apoptosis of epithelial enterocytes, immune activation in the gut, recruitment of innate immune cells and inflammation, leading to further damage of the gut epithelium fueling mucosal barrier destruction and microbial translocation that trigger chronic systemic immune activation and inflammation [66,68,108].

Given the importance of the initial virus-host interactions and viral cytopathic effects in establishing this vicious cycle, we assessed the dynamics of viral replication in the GI tract after intrarectal challenge and found high levels of viral replication starting as early as 4–6 dpi in the gut and even earlier in the draining LNs. We anticipated that these high levels of viral replication would be paralleled by a depletion of CD4+ T target cells. Yet, in spite of the high levels of

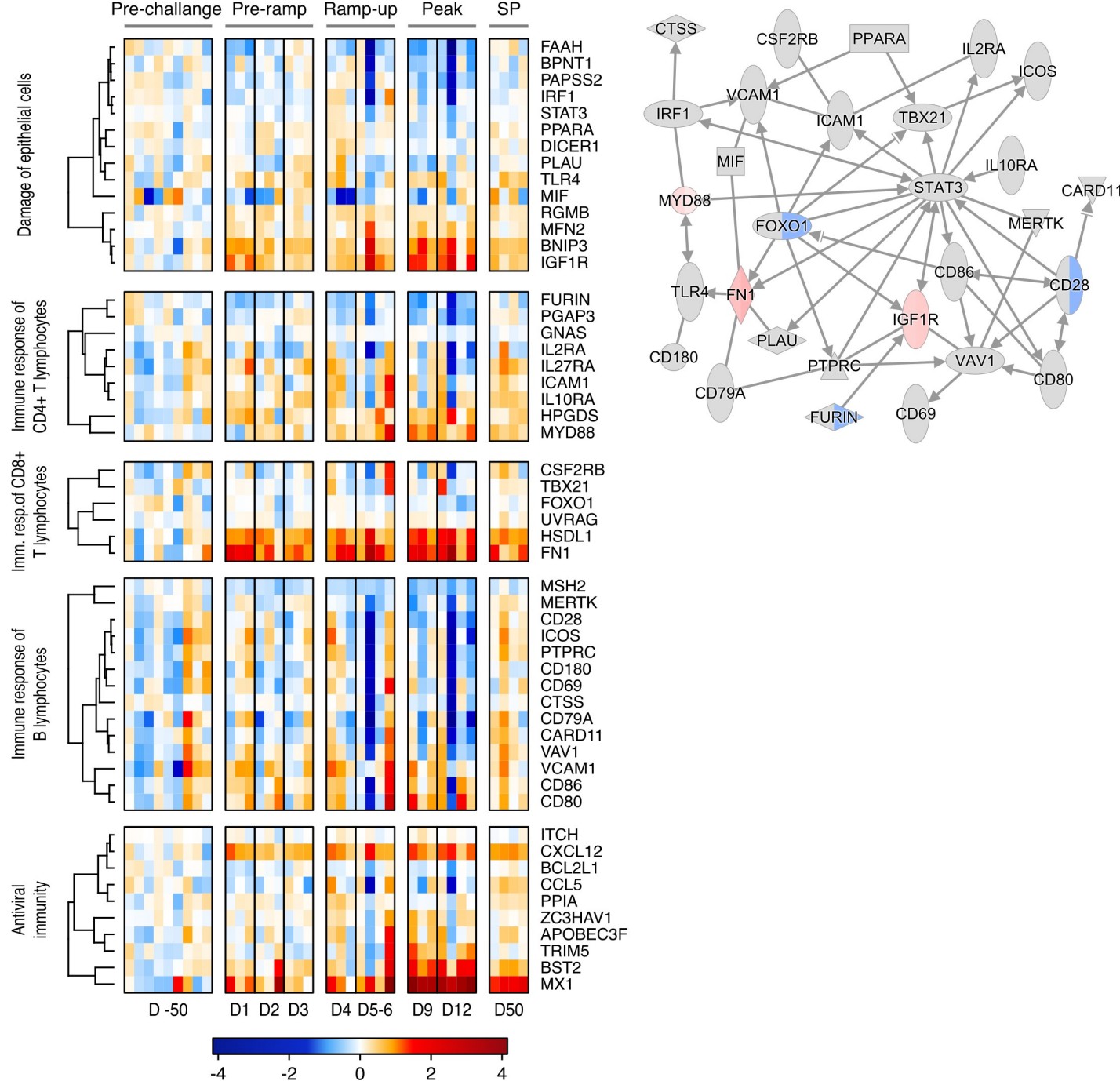

**Fig 9. Heatmap & network of gene expression in the gut during SIV infection in AGMs.** RNAseq data from AGM gut tissue displayed as (A) a heatmap showing gene expression changes in genes associated with biological processes related to SIVsab infection, immune responses and damage to the gut epithelium. Here, pre-challenge represents the changes in gene expression between each of the individual animals preinfection versus the average changes for the entire pre-challenge group in order to show natural genetic variation. The changes in pre-ramp (1–3 dpi), ramp-up (4–6 dpi), peak (9–12 dpi) and set-point (SP, 46–55 dpi) groups represent the change in gene expression post-infection for each animal compared to the pre-challenge group averages. (B) a network showing direct interactions between genes in SIV related processes at the peak of infection (9–12 dpi). For the heatmap, the level of alteration of gene expression is shown in blue (downregulation) and red (upregulation), with genes clustered using a Spearman correlation, with the dendrogram showing relationship displayed on the left. For the pathway, each gene is shown as a node and the arrows between nodes indicate direct gene interactions. Red nodes indicate upregulation of gene expression, green nodes indicate downregulation. In split nodes, the left and right node sections correspond to 9 dpi and 12 dpi, respectively.

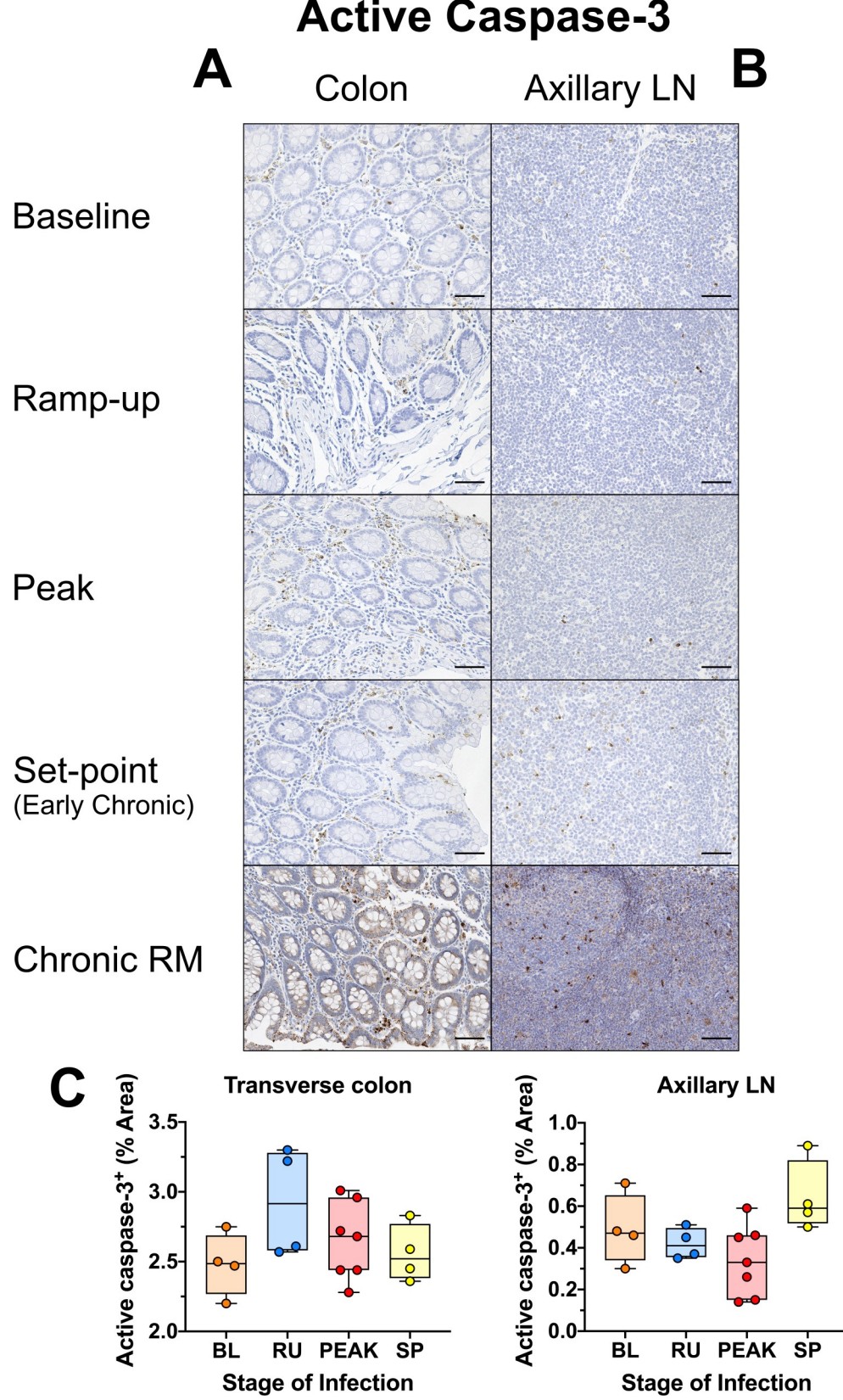

**Fig 10. Immunohistochemistry (IHC) for apoptosis in SIVsab-infected AGMs.** DAB-based IHC for active caspase-3 in the (A) transverse colon, and (B) axillary LN of AGMs and chronically SIV-infected rhesus macaques (RMs). In all the images, positive DAB signal is shown in brown, with the remaining tissue counterstained blue. The representative images display lateral cuts of the crypts to show apoptosis in both the epithelium and lamina propria. Below are shown image quantifications (C) of the percent area of the total positive DAB signal. The quantification for each animal represents the average of the values from 9–12 individual image quantifications. The four time groups are based on the days postinfection, with: BL (baseline, preinfection, orange), PRU (preramp, 1–3 dpi, green) RU (ramp-up, 4–6 dpi, blue), PEAK (peak, 9-12dpi, red) and SP (set-point, 46–55 dpi, yellow). All AGM quantifications were performed using FIJI version 1.0. Asterisks indicate statistical significance, with $^* = p < 0.05$. All AGM images were captured at 200X magnification using an AxioImager M1 bright-field microscope equipped with an AxioCam MRc5. Scale bar: 100 μm.

virus replication in both the gut and LNs, the overall levels of CD4$^+$ T cells were relatively stable during the early stages of infection, in contrast to previous reports of significant mucosal CD4$^+$ T cell depletion in AGMs during acute and early chronic infection [22,101,102]. One possible explanation for this discrepancy is that the previous studies relied on longitudinal blood and tissue sampling from the same animal, differently from our cross-sectional study which compared data among different animals. Individual variations between animals may thus have partially obfuscated the CD4$^+$ depletion. Indeed, for the blood and LNs, where preinfection samples were available for all the animals, allowing for direct pre- *versus* postinfection comparisons, we were able to document CD4$^+$ T cell depletion.

As early viral replication induces high levels of immune activation in pathogenic infections [4], we assessed the levels of immune activation in early SIVsab infection of AGMs. As a marker of activation via proliferation, Ki-67 expression by CD4$^+$ and CD8$^+$ T cells or CD20$^+$ B cells did not significantly increase in either the gut or the LNs at any time, by either flow cytometry or IHC. While low levels of chronic immune activation are in agreement with previous studies of natural hosts [22,23,101,105,109], increased acute T-cell activation in the LNs was reported for sooty mangabeys [110], suggesting that reduced early immune activation may be characteristic to AGMs. The observed low levels of immune activation are also supported by the very few alterations in gene expression that would indicate a major uptick in CD4$^+$ or CD8$^+$ T-cell activation during acute infection and essentially no major alteration to CD20$^+$ linked genes. The most notable alterations occurred around the peak of infection and were linked to interferon-stimulated genes (ISGs), in agreement with previous studies [56].

As T cell immune activation is usually accompanied by an innate immune response, we assessed neutrophil recruitment to the gut and found little to no neutrophil accumulation in the gut mucosa and superficial LNs. These results are in stark contrast with the large scale recruitment of MPO-positive neutrophils to the gut epithelium in chronically SIV-infected RMs [62], HIV-infected subjects [53], or in other pathological conditions associated with gut inflammation and microbial translocation, such as IBD [111]. We also saw no major change in MPO gene expression levels in the AGMs at any time during infection, supporting the lack of sustained alteration to neutrophil populations in response to SIV infection in AGMs.

Interestingly, the AGMs actually had similar levels of neutrophils in the gut to some of the chronically infected RMs and to RMs during early acute infection [62]. Historically, it has been shown that both uninfected RMs [73] and HIV-uninfected patients [53,112] tend to have very small numbers of neutrophils resident in the gut, so their presence in AGMs suggests some role in responding to SIV infection. One possible explanation is that the resident gut neutrophils respond immediately to any possible leakage, no matter how minimal, from the gut lumen during the acute infection, through mechanisms like phagocytosis and the release of NETs [113,114]. Indeed, we did see a significant increase in the number of MPO-positive cells in the colon only during the ramp-up period, possibly indicating a transient increase of neutrophils during the earliest period following infection. However, any activity by neutrophils

# Claudin-3

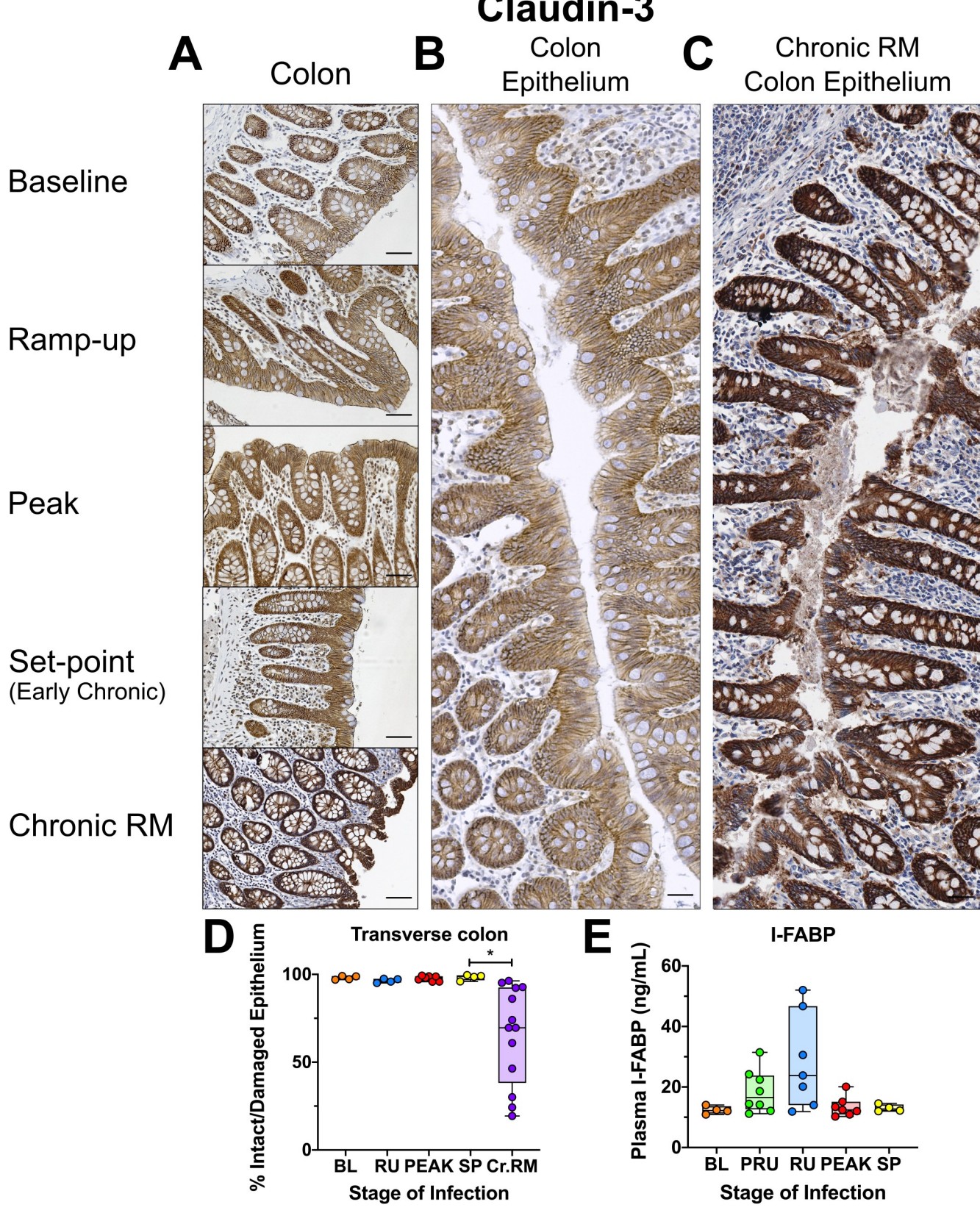

**Fig 11. Immunohistochemistry (IHC) for mucosal integrity and continuity in SIVsab-infected AGMs.** DAB-based IHC for claudin-3 in the transverse colon of AGMs and chronically SIV-infected rhesus macaques (RMs). In all images, positive DAB signal is shown in brown, with the remaining tissue counterstained blue. The representative images of the gut display longitude cuts of the villi to show continuity of the epithelial barrier (A). (B) Composite image of continuous transverse colon epithelium from an AGM during peak acute viral replication. (C) Image of continuous transverse colon epithelium from an RM during late chronic viral replication. (D) The percent area of the total positive DAB signal and the ratio of damaged/total epithelium. (E) Plasma concentrations in ng/μL of I-FABP, a marker of epithelial damage in the gut. The images were stitched together to form a composite image using the Stitching plugin for FIJI version 1.0. Overall continuity of the mucosal epithelium was estimated by tracing the length intact vs damaged epithelium and quantifying their relative areas. The total DAB quantifications for the AGMs represent the average of the values from 9–12 individual image quantifications, while the epithelial damage quantifications represent 3–4 composite images. The four different time groups are based on the days post infection, with: BL (baseline, preinfection, orange), PRU (preramp, 1–3 dpi, green) RU (ramp-up, 4–6 dpi, blue), PEAK (peak, 9-12dpi, red) and SP (set-point, 46–55 dpi, yellow). The chronic RMs are shown in purple. All AGM quantifications were performed using FIJI version 1.0. Asterisks indicate statistical significance, with ** = $p<0.01$. All AGM individual images were captured at 200X magnification using an AxioImager M1 bright-field microscope equipped with an AxioCam MRc5. For the composite images, each individual image was captured at 100X magnification. Scale bar: 100 μm.

would have to be carefully controlled to minimize collateral damage to the epithelium, as occurs in many inflammatory mucosal disorders [115]. Another possible explanation for their presence is that, in AGMs, the resident gut neutrophils play a nontraditional role, directly contributing to maintaining the epithelial barrier, though the vast majority of evidence supports neutrophils having net negative impact on tissue homeostasis [116].

Previous research has shown that in both pathogenic and nonpathogenic hosts, activated immune cells and epithelial cells exhibit increased production of proinflammatory cytokines in response to acute HIV/SIV infection [3,62,117–119]. However, these high levels of inflammation are sustained chronically only in pathogenic hosts, where they are strong predictors of progression to AIDS and death [3,18,30,54,110,120], while being resolved in nonpathogenic hosts by the rapid establishment of an anti-inflammatory milieu [48]. Accordingly, only limited and transient inflammation occurred in the AGMs, which was mostly resolved by chronic infection [2,3,28,48,56,57]. Additionally, immediately following infection the AGMs exhibited increased expression of the gene encoding IL-17B, a T cell-derived cytokine with potential anti-inflammatory effects in the gut [91].

However, control of inflammation does not mean lack of response by the host, as previously demonstrated by the ability of natural hosts to mount robust innate immune responses during acute infection [41,55,56,121], particularly the type 1 ISGs genes (ISGs), including MX1, MX2 and IP-10. These responses are then rapidly resolved at the transition to chronic infection, as observed here with MX1 in the AGMs. Conversely, SIV-infected RMs exhibit strong innate immune activation and ISG upregulation throughout the entire course of SIV infection [41,55,56,121], as reflected by the high levels of MX1 in RMs. Along with MX1, we also observed increases in the expression of the ISGs CXCL10 (IP-10) and CXCL11 (I-TAC) at the peak of viral replication in both AGMs and RMs. These increases were resolved only in AGMs and persisted in the RMs. Persistence of IFN-based responses in pathogenic SIV infections eventually induces IFN desensitization and decreased ISG expression, resulting in increased SIV reservoir size and CD4[+] T cell loss [84].

Interestingly, while some acute SIV infection studies in natural hosts observed upregulation of *MX1* and other ISGs as early as 1 and 6 dpi [56], here the tissue protein levels of MX1 did not increase prior to the peak. This discrepancy may be due to the route of inoculation (mucosal *versus* IV), that could shift the acute infection towards later time points [22]. It is also possible there was an "inoculum effect", where introduction of large amount virus directly into the bloodstream stimulated ISGs immediately. Additionally, some of the individual AGMs did exhibit increased expression levels of ISGs, including MX1, earlier during infection. This suggests that the early virus replication might be sufficient in some cases to induce a response that is both transient and of a far lesser magnitude than seen at the peak of virus replication.

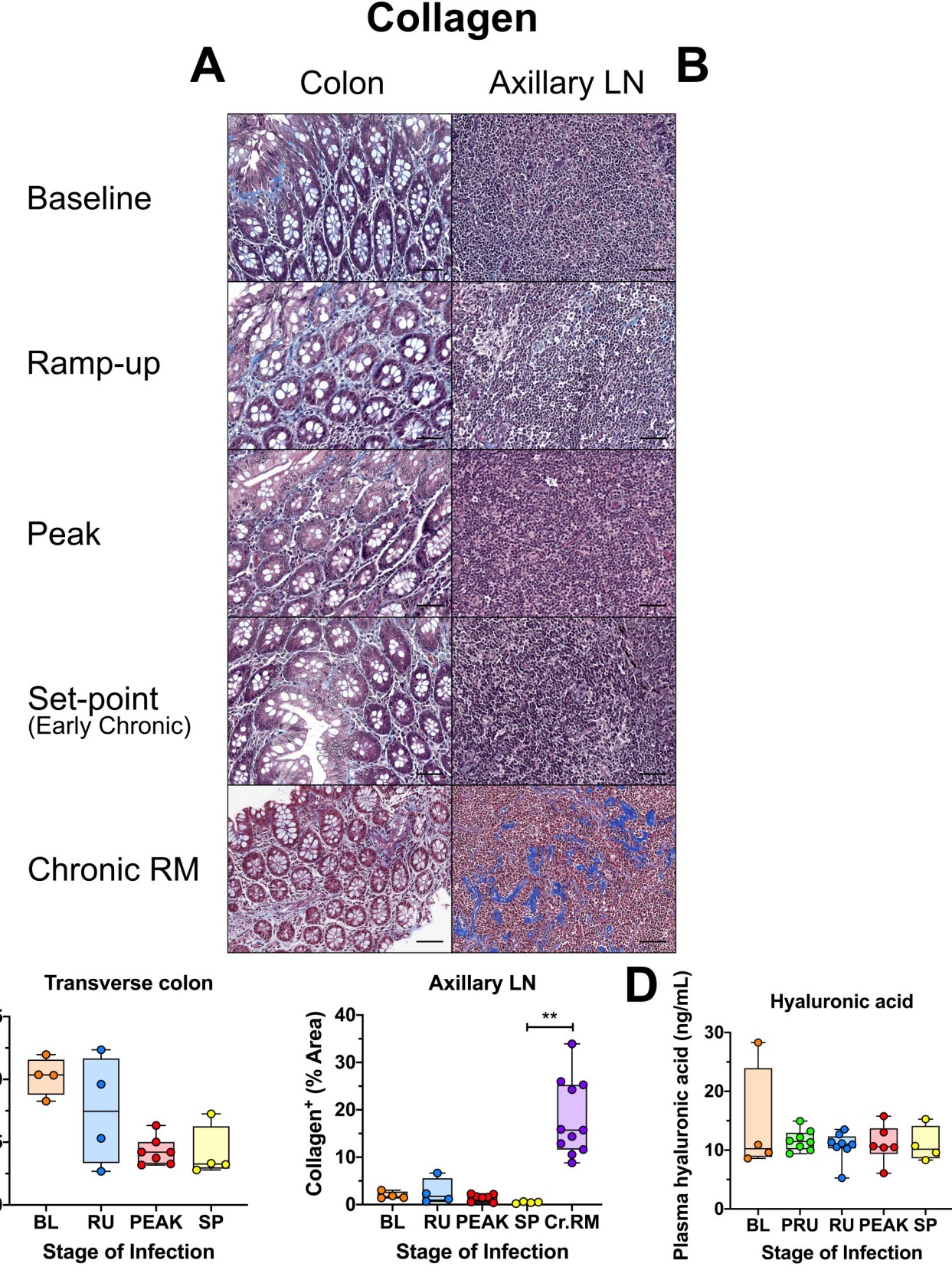

**Fig 12. Immunohistochemistry (IHC) for collagen deposition and fibrosis in SIVsab-infected AGMs.** Masson's trichrome stain for collagen in the (A) transverse colon, and (B) axillary LN of AGMs and chronically SIV-infected rhesus macaques (RMs). For the trichrome stain, red/pink are myosin fibers and cytoplasm, black is nuclei, and blue is collagen. The representative images of the gut display longitude cuts of the villi to better show the presence of collagen in the lamina propria. Below are shown image quantifications (C) of the percent area of the total blue signal indicating collagen. (D) Plasma concentrations in ng/µL of hyaluronic acid, a marker of fibrosis. The quantification for each animal represents the average of the values from 9–12 individual image quantifications. The four different time groups are based on the days post infection, with: BL (baseline, preinfection, orange), PRU (preramp, 1–3 dpi, green) RU (ramp-up, 4–6 dpi, blue), PEAK (peak, 9-12dpi, red) and SP (set-point, 46–55 dpi, yellow). The chronic RMs are shown in purple. All AGM quantifications were performed using FIJI version 1.0. Asterisks indicate statistical significance, with $^{**} = p{<}0.01$. All AGM images were captured at 200X magnification using an AxioImager M1 bright-field microscope equipped with an AxioCam MRc5. Scale bar: 100 µm.

It should also be noted that there was an apparent discrepancy between the MX1 levels observed in the IHC staining versus the gene expression levels; namely, that the gene expression levels of MX1 appear to remain elevated into set-point of infection, while the tissue levels appear to return to baseline. This may largely be due to way the data is displayed. At viral set point, MX1 is significantly higher than pre-challenge (*p = 0.0082*), but not after adjustment for multiple comparisons (adjusted *p = 0.082*). The log2 fold change is 2.06. By comparison, at D9 and D12, the log2 fold change is 6.96 and 7.34, respectively. The color scale in the Figure is balanced to show the numeric range with the most variation (-1.5 to 1.5), which is why the high values look similar. Therefore, we feel that there are likely still cells expressing the MX1 gene which are present at by the set-point of viral replication, but not enough to result in widespread MX1 tissue expression.

Apart from the ISGs, we also observed up-regulation of several genes linked to T-cell activation. One of these, HPGDS, can be linked to proinflammatory Th2 CD4$^+$ T cells. However, this primarily occurs in eosinophil-driven allergic responses in the gut [122]. We also observed up-regulation of MYD88, an essential TLR signal transducer. MYD88 has been shown to be important for regulation of CD4$^+$ T cells, including in promoting activation in the murine inflammatory bowel disease model [123]. As such, MYD88 could potentially be directly associated with an antimicrobial response and microbial translocation. However, MYD88 has also been linked to promotion of wound healing responses through TLR signaling [124,125]. However, up-regulation of both of these genes was relatively low.

In contrast, we found much stronger up-regulation of HSDL1 and FN1 at the peak of viral replication. HSDL1 is largely nonfunctional [126], but FN1 has been shown to activate CD8$^+$ T cell degranulation [127]. We recently reported FN1 as a key part of the wound healing pathway in AGMs. Early repair of virally-induced mucosal damage could contribute to the resolution of inflammation during chronic SIV infection of natural hosts [86].

The tissue repair response might be associated with STAT3, which, in addition of being associated with regulation of proinflammatory cytokines and chemokines, is also critical to the signaling necessary for repair of damage to the gut [128–130].

In pathogenic infections, increased epithelial cell death [93] and loss of mucosal barrier integrity result in high levels of proliferation and turnover, and degradation of the tight junctions between the epithelial cells [62,131]. Tight junctions are vital not only for cell-to-cell adhesion, but also for regulating paracellular permeability and allowing small molecules and ions to move across the epithelial barrier. Any loss of tight junction proteins (i.e., claudin-3), due to cell death or internalization *via* endocytosis in response to infection would be indicative of a loss of mucosal integrity [60]. In SIV-infected RMs, claudin-3 staining has been used to measure the breakdown in mucosal epithelium continuity, while expression of claudin-encoding genes (i.e., *CLDN3*) has been shown to be downregulated as early as 3 dpi [132], similar to experimentally-induced IBD [95]. Importantly, at no timepoint post SIVsab infection in AGMs, did we find increases in mucosal apoptosis, proliferation or loss of the mucosal epithelium integrity, in stark contrast with reduced mucosal epithelium integrity in chronically

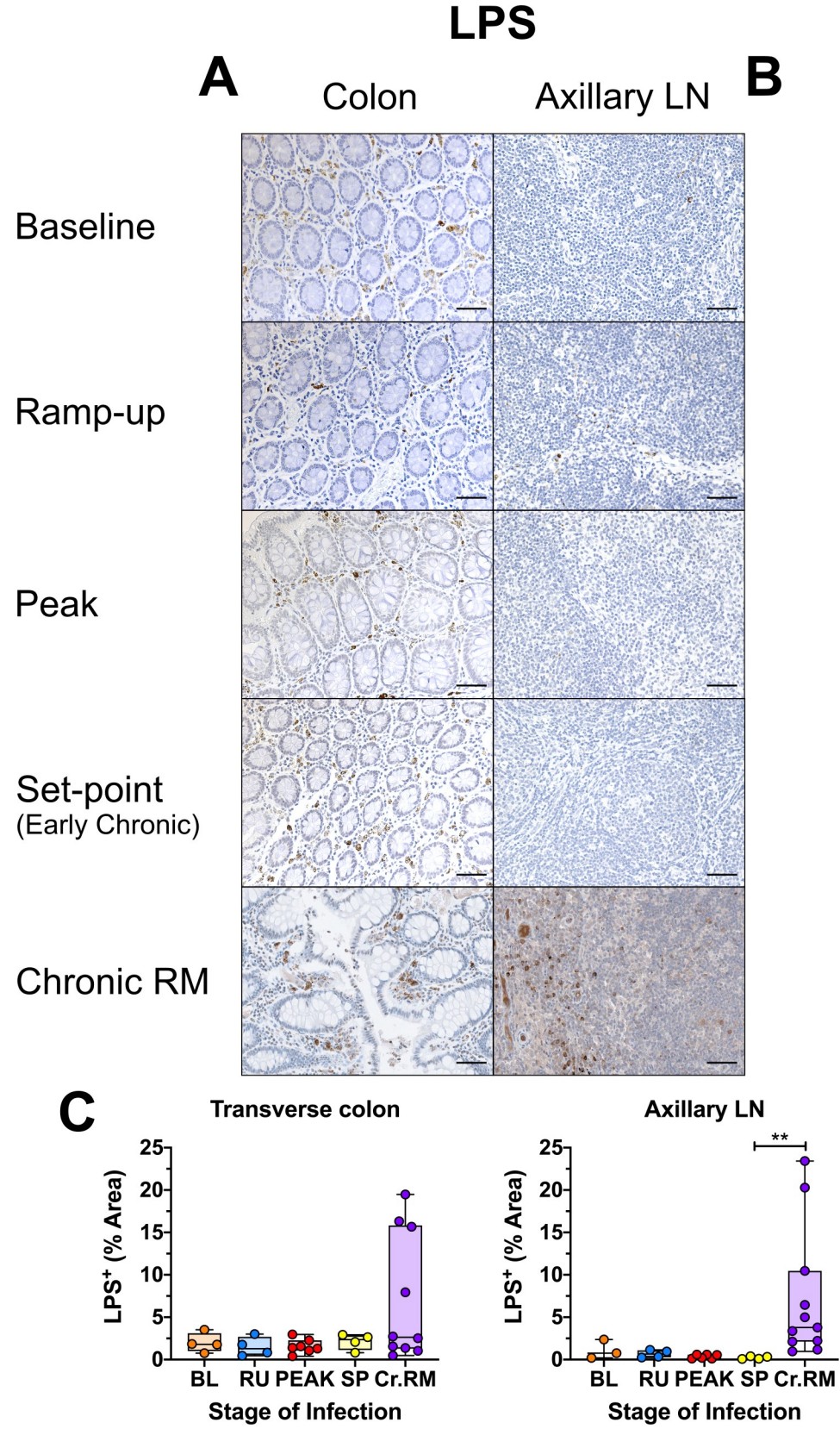

**Fig 13. Immunohistochemistry (IHC) for the translocation of LPS in SIVsab-infected AGMs.** DAB-based IHC for LPS-core protein in the (A) transverse colon, and (B) axillary LN of AGMs and chronically SIV-infected rhesus macaques (RMs). In all the images, positive DAB signal is shown in brown, with the remaining tissue counterstained blue. Below are shown image quantifications (C) of the percent area of the total positive DAB signal. The quantification for each animal represents the average of the values from 9–12 individual image quantifications. The four time groups are based on the days postinfection, with: BL (baseline, preinfection, orange), PRU (preramp, 1–3 dpi, green) RU (ramp-up, 4–6 dpi, blue), PEAK (peak, 9-12dpi, red) and SP (set-point, 46–55 dpi, yellow). The chronic RMs are shown in purple. All AGM quantifications were performed using FIJI version 1.0. Asterisks indicate statistical significance, with ** = $p<0.01$. All AGM images were captured at 200X magnification using an AxioImager M1 bright-field microscope equipped with an AxioCam MRc5. Scale bar: 100 μm.

infected RMs [62,73]. The histological data were supported by either flow cytometry or measurements of biomarkers of gut integrity (such as I-FABP or hyaluronic acid) in the plasma. We also observed upregulation of several genes linked to wound healing and repair of epithelial damage, suggesting that the lack of damage might be connected to maintenance of the gut epithelium itself, as reported [86]. The importance of maintenance of the gut barrier in natural hosts was shown when experimental colitis induced in AGMs recapitulated many of the features of pathogenic SIV infections [73].

Apart from loss of gut mucosal integrity, increased collagen deposition resulting in rapid destruction of LN architecture and fibrosis is also a hallmark of pathogenic HIV/SIV infections and impaired immune recovery in the gut [92,133]. We found no evidence of increased fibrosis (i.e., collagen deposition) in either the gut or LNs of SIVsab-infected AGMs, and we confirmed this result by showing stable levels of hyaluronic acid in the plasma throughout the follow-up. As limiting fibrosis and maintaining the LN architecture has been shown to be critical in preserving CD4+ T cell populations during pathogenic SIV infection [134], avoiding early fibrosis in the LNs could contribute to prevention of disease progression in AGMs.

Together, our results demonstrate that acute SIV infection of AGMs is associated with no increase in intestinal epithelial apoptosis, fibrosis, or breakage of the mucosal epithelium, supporting the notion that the overall integrity of the intestinal epithelium is maintained throughout the course of SIV infection.

Altogether, we could not document any increase in microbial translocation throughout the acute and early chronic SIVsab infection AGMs by either quantitative IHC or serology, in contrast with SIV-infected RMs, HIV-infected subjects, or SIV-infected AGMs with experimental colitis [51,59,62,73,81]. It is worth noting that while a significant decrease in plasma levels of sCD14 were observed, this decrease was only transient. Furthermore, sCD14 has been shown to be directly correlated with LPS levels during HIV/SIV infection [51,69,78]. Conversely, we previously showed that in pathogenic hosts therapeutic reduction in microbial translocation resulted in lower plasma levels of sCD14 [67,74]. Therefore, thought the exact cause of this transient decrease in sCD14 is unclear, it is unlikely to be associated with any increase in microbial translocation.

Interestingly, despite the lack of microbial translocation, higher baseline levels of LPS, but not of *E. coli* were observed in the lamina propria of the transverse colon of SIV-uninfected AGMs compared to SIV-uninfected RMs, though these levels were stable even after infection. It is possible that gut resident phagocytic cells in AGMs, such as neutrophils, macrophages or dendritic cells, take up LPS from the colonic lumen, but actively attenuate inflammatory responses. Human colonic macrophages are highly anergic to LPS as a proinflammatory stimuli, possibly through interaction with the commensal colonic flora [135,136], making it plausible that an analogous process of anergy to LPS may exist in AGMs. This may even contribute to dampening the early inflammatory response induced by microbial translocation following SIV infection. Indeed, SMs, another natural SIV host have an alteration to the TLR4 gene

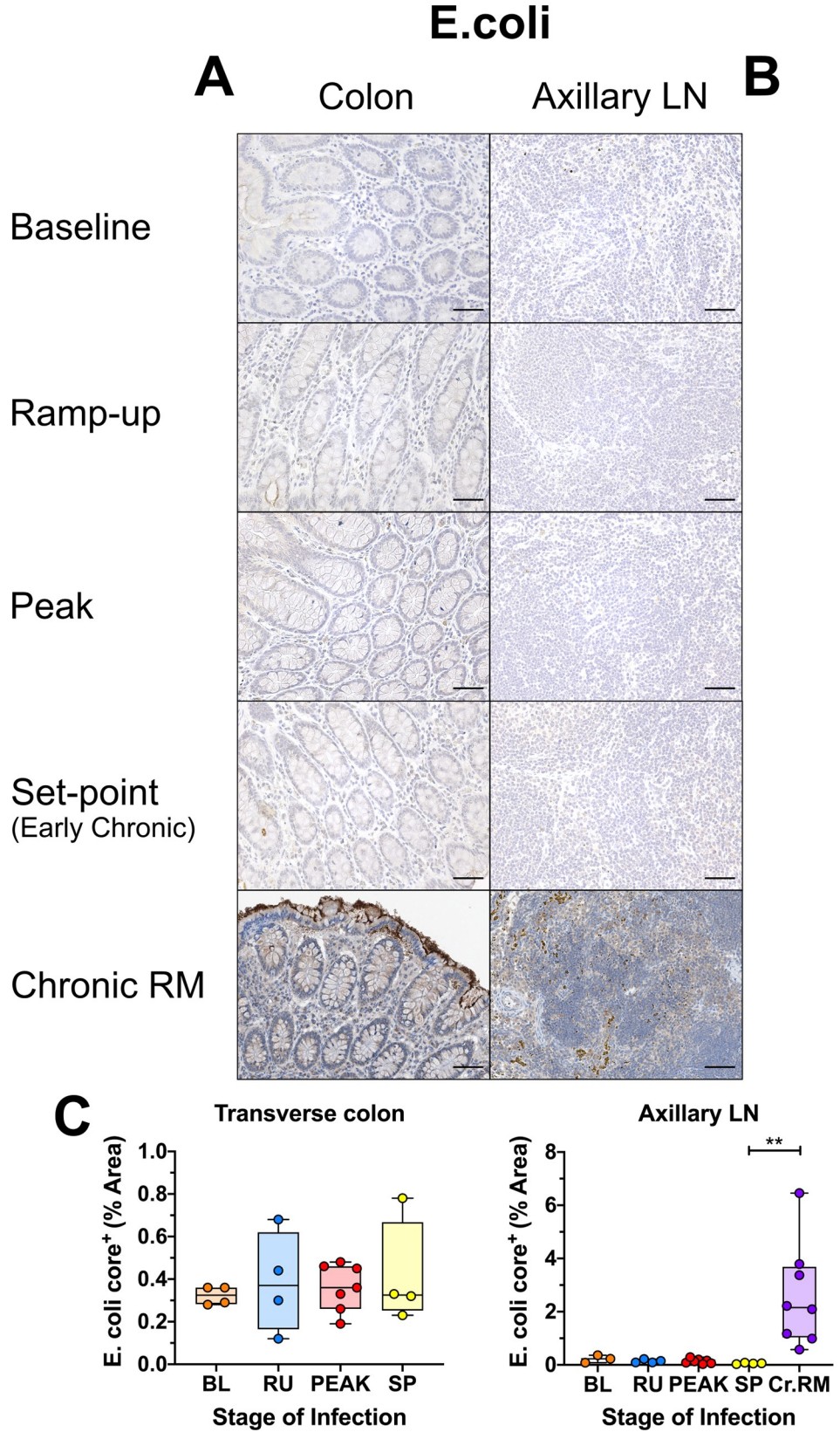

**Fig 14. Immunohistochemistry (IHC) for translocation of *E. coli* in SIVsab-infected AGMs.** DAB-based IHC for *E. coli* in the (A) transverse colon, and (B) axillary LN of AGMs and chronically SIV-infected rhesus macaques (RMs). In all the images, positive DAB signal is shown in brown, with the remaining tissue counterstained blue. Below are shown image quantifications (C) of the percent area of the total positive DAB signal. The quantification for each animal represents the average of the values from 9–12 individual image quantifications. The four different time groups are based on the days post infection, with: BL (baseline, preinfection, orange), PRU (preramp, 1–3 dpi, green) RU (ramp-up, 4–6 dpi, blue), PEAK (peak, 9-12dpi, red) and SP (set-point, 46–55 dpi, yellow). The chronic RMs are shown in purple. All AGM quantifications were performed using FIJI version 1.0. Asterisks indicate statistical significance, with ** = $p<0.01$. All AGM images were captured at 200X magnification using an AxioImager M1 bright-field microscope equipped with an AxioCam MRc5. Scale bar: 100 μm.

associated with a blunted response to TLR4 ligands [38]. As LPS is an important TLR4 ligand, a similar LPS anergy in AGMs could promote an attenuated SIV pathogenesis.

Additionally, LPS-core was almost completely absent from the LNs in the AGMs throughout the follow-up, in sharp contrast with the chronically SIV-infected RMs, where LPS-core was present in the LNs of some animals at high levels. This suggests that the LPS found in the lamina propria of the colon in AGMs is retained within the GI tract and not systemically disseminated, in contrast to SIV-infected RMs.

Our results thus show that, in the absence of mucosal damage, the transient increases in inflammatory responses observed in natural SIV infection do not have any major pathogenic consequence. It follows that any interventions to prevent or modulate intestinal mucosal damage during HIV/SIV pathogenic infection may prevent disease progression, even in the presence of active viral replication. A similar situation has been observed with viremic nonprogressor HIV-infected subjects, where limited immune activation and inflammation associate a lack of disease progression [137,138].

In conclusion, we showed that SIVsab-infected AGMs maintain gut mucosa integrity throughout infection, and lack any of the indicators of damage or breakage of the gut epithelium that are seen in chronically SIV-infected RMs. We also failed to document any increase in microbes or microbial byproducts in the gut and the LNs, indicating a total absence of microbial translocation during SIV infection in AGMs. We observed only transient immune activation and a bipartite wave of proinflammatory cytokine production during the acute infection and little to no increase in the markers of the aberrant chronic immune activation and inflammation typical of pathogenic HIV/SIV infections. We also confirmed that there are few alterations in the expression of genes linked to immune activation, inflammation or epithelial damage in the AGMs. As such, we directly demonstrated for the first time that the lack of gut dysfunction, damage to the gut epithelium and the subsequent microbial translocation during early SIV infection are important factors behind the control of chronic immune activation and inflammation, and thus lack of disease progression in the natural hosts of SIVs, though other factors, such as TLR4 expression or virus tropism, might influence disease progression in other African NHP species [2,9,26,38,42,47,107]. Our study suggests early interventions aimed at repairing or preventing gut epithelial damage may represent a viable alternative to the current interventions intended to curb the deleterious consequences of HIV infection.

## Materials and methods

### Ethics statement

All the AGMs included in this study were housed at the RIDC animal facility of the University of Pittsburgh, an AAALAC International facility, as per the regulations outlined in the *Guide for the Care and Use of Laboratory Animals* [139] and the Animal Welfare Act [140]. All animal experiments were approved by the University of Pittsburgh Institutional Animal Care and Use

Committee (IACUC). Efforts were made to minimize NHP suffering, in agreement with the recommendations in "*The Use of Nonhuman Primates in Research*" [141]. The NHP facility was air-conditioned, with an ambient temperature of 21–25˚C, a relative humidity of 40–60% and a 12-hour light/dark cycle. AGMs were socially housed in suspended stainless-steel wire-bottomed cages. A variety of environmental enrichment strategies were employed, including providing toys to manipulate and playing entertainment videos in the NHP rooms. The NHPs were observed twice daily and any signs of disease or discomfort were reported to the veterinary staff for evaluation. At the completion of the study, the NHPs were euthanized following procedures approved in the IACUC protocol (#1008829).

## Animals & infections

Thirty-three adult (4–9 years old) male AGMs (*Chlorocebus sabaeus*) of Caribbean origin were included (S1 Table). They were intrarectally (ir) challenged with infectious plasma originally collected from an acutely infected AGM, diluted to contain $10^7$ RNA copies of SIVsab [20]. This dosage was established during a preliminary study as a relatively low dose that could reliably infect adult AGMs intrarectally [22]. Infectious acute plasma was used as a highly infectious inoculum that better reflects viral diversity in the wild [142]. Male AGMs and the ir route of infection were preferred to avoid the physiological variability of the vaginal mucosa associated with the estrus cycle [143] even though vaginal transmission is more prevalent in wild NHPs.

The AGMs were euthanized serially throughout the acute and early chronic SIV infection and were divided into the following groups based on their predicted viremic status at the time of sacrifice: (i) preinfection (baseline); (ii) preramp [1–3 days postinfection (dpi)]; (iii) ramp-up (4–6 dpi); (iv) peak (9–12 dpi); (v) set-point (46–55 dpi).

For the purposes for comparison to a pathogenic SIV infection, we included results from chronically SIVmac-infected adult male RMs sacrificed during chronic infection (127–410 dpi). References for the care and infection strategies used for the RMs are listed in S2 Table.

## Tissue sampling & isolation of mononuclear cells

Blood, intestinal and LN biopsies were collected from each AGM prior to infection to establish the baseline levels for the tested markers. Then, immediately prior to necropsy, a maximum bleed was performed, and EDTA, heparin and sodium citrate blood were collected. An extensive tissue sampling was performed during the necropsy. Sections of numerous tissue sites were collected for snap freezing in liquid nitrogen, or fixation in 4% paraformaldehye. Additional samples from tissues of interest were collected for cell separation, including the jejunum, transverse colon and axillary LNs. Cells were separated from whole blood and tissues and used fresh for flow cytometry. Any remaining cells were frozen at -80˚C for later use in freezing medium containing heat-inactivated fetal bovine serum with 10% dimethyl sulfoxide (DMSO).

Plasma was separated from whole blood within 1 hr of collection by centrifugation at 2,200 rpms for 20 min, aliquoted and snap frozen at -80˚C. Peripheral blood mononuclear cells (PBMCs) were then isolated and frozen as described [102,144].

Cells were separated from LNs by first mincing the tissue and then pressing it through a 70 μm nylon filter, as described [23,145]. LN biopsies were also dry frozen at -80˚C for DNA/RNA extraction or fixed in 4% paraformaldehyde for histological processing.

The intestinal biopsies were fixed and frozen in a similar fashion, though they were preserved in RNALater (Thermo Fisher) for use in RNAseq instead of 4% paraformaldehyde for histological processing. No cells were separated from the intestinal biopsies due to their small

size and the need to prioritize other assays. Cells from the much larger gut sections taken at necropsy were isolated as previously described [22,75]. Briefly, the sections were first trimmed of fat, then opened longitudinally and gently scraped to remove waste. Next, the gut sections were diced into ~1-2mm$^2$ pieces, incubated twice in HBSS (Lonza) with 8 mM EDTA (Fisher Scientific, Pittsburgh, PA) for 30 min. at 37˚C while shaking at 300 rpms on an orbital shaker/incubator, digested twice with RPMI with 0.75% collagenase (Sigma-Aldrich, St. Louis, MO) under the same conditions and separated by density gradient centrifugation at 2,200 rpms for 20 min using 35% and 60% percoll (Sigma-Aldrich), as described [24,102,146].

The tissues from the RMs were collected and processed in a similar manner to the AGMs, as previously described [62,110].

### Plasma viral RNA extraction

Viral RNA was extracted from plasma using the QIAGEN viral RNA Mini kit (QIAGEN, Germantown, MD). RNA was then eluted and reverse transcription was preformed using the Taqman Gold reverse transcription PCR (RT-PCR) kit and random hexamers (PE, Foster City, CA), as described [20]. The RT-PCRs were run in a Gene Amp PCR System 9700 thermocycler (Applied Biosystems, Grand Island, NY).

### Tissue DNA/RNA extraction

DNA/RNA was extracted from the liquid nitrogen snap frozen tissue sections as follows: the sections were carefully moved on dry ice from standard cryotubes to SPEX SamplePrep polycarbonate cryotubes. Then, stainless steel ball bearings and hex nuts (MSC, St. Louis, MO) were layered over each tissue before TriReagent (Molecular Research Center, Cincinnati, OH) was added at an approximately 100 mg/1 mL ratio. The tissues were homogenized using a SPEX Geno/Grinder (SPEX SamplePrep, Metuchan, NJ), with a 1–2 cycles of agitation at 1,600 rpms for 2 min. Following homogenization, the tissues were allowed to sit for 15–20 min in TriReagent to solubilize small tissue particles. Next, 1 mL of tissue lysate was transferred from each sample to 1.5 mL snap cap tube. The RNA was extracted first by adding 100 μL bromochloropropane (MRC, Cincinnati, OH) to each sample, vortexing for 30 sec, and then spinning at 14,000xg for 15 min at 4˚C. After centrifugation, the upper aqueous phase was removed and transferred to a tube with 12 μL of 20 mg/mL glycogen (Sigma-Aldrich) before being mixed with 500 μL isopropanol. The solution was then spun at 21,000xg for 10 min at room temperature (RT), the supernatant was removed, and the pellet was washed with 600–800 μL 70% ethanol. Each pellet was allowed to sit under ethanol for 3 days at -20˚C before being dried and either immediately resuspended or stored at -80˚C for future use. The same protocol was repeated for DNA, except with 500 μL DNA Back Extraction solution (4 M GuSCN, 1 M Tris base, 50 mM sodium citrate) instead of BCP.

### Plasma VLs quantification

Plasma VLs were monitored by real time quantitative PCR (qPCR) based on a 180-bp segment located in the *gag* region, as described [48,101,147], using primers and probes specifically designed for SIVsab92018 [101,102] were synthesized by Integrated DNA Technologies (Coralville, IA). All qPCRs were run in duplicate and negative controls for the RT-PCR and qPCR were included on each plate. The qPCRs were run on 7900HT Fast Real Time System (Applied Biosystems), as described [101].

qPCR was performed by using TaqMan® Gene Expression Master Mix (PE Applied Biosystems), as described [148,149], using previously published SIVsab-specific primers and probed [35,86].

Absolute viral RNA (vRNA) copy numbers were calculated relative to amplification of an SIVsab standard, which was subjected to RT-PCR in parallel with the samples being tested. The standard was generated as described [101]. The detection limit of this conventional qPCR was 30 vRNA copies/mL of plasma [101].

## Tissue VLs quantification

Tissue VLs were quantified by real time quantitative PCRs (RT-qPCRs) based on a 180-bp segment located in the *gag* region [48,101,147] and the TaqMan® Gene Expression Master Mix (PE Applied Biosystems). Primers and probes [101,102] were synthesized by Integrated DNA Technologies (Coralville, IA). All qPCRs were run in duplicate and negative controls for the RT-PCR and qPCR were included on each plate. The qPCRs were run on 7900HT Fast Real Time System (Applied Biosystems), as described [101].

Absolute viral RNA and DNA (vRNA/vDNA) copy numbers were calculated relative to amplification of an SIVsab standard, which was subjected to RT-PCR in parallel with the samples being tested. The standard was generated as described [101]. To quantify the number of vRNA/vDNA copies per million somatic cells, we also ran the samples with primers and probes for RM CCR5, as follows: RM-CCR5-F (5'-CCA-GAA-GAG-CTG-CGA-CAT-CC-3'), RM-CCR5-R (5'-GTT-AAG-GCT-TTT-ACT-CAT-CTC-AGA-AGC-TAA-C-3', RM-CCR5-Probe (5'-CalRed610-TTC-CCC-TAC-AAG-AAA-CTC-TCC-CCG-GTA-AGT-A-BHQ2-3').

## Flow cytometry

Cells were stained for flow cytometry as described [102,145]. Briefly, whole blood was lysed using fluorescence-activated cell sorter (FACS) lysing solution (BD Biosciences). Lysed blood and isolated immune cells from LNs and intestine were incubated at 4°C for 30 min with monoclonal antibodies (mAbs). Cells were then washed with phosphate-buffered saline (1x PBS) and then fixed with a BD stabilizing fixative (BD Bioscience). For intracellular stains, after the surface stain was completed, the cells were fixed with a 4% paraformaldehyde solution for 20 min. Cells were then washed with 1x PBS, followed by a wash with 0.1% saponin solution and another mAbs incubation. Cells were washed with 0.1% saponin solution and fixed with a BD stabilizing fixative. The absolute counts of peripheral blood lymphocytes were determined by using TruCount tubes (BD Bioscience) [148,150]. First, blood CD45[+] cells were quantified using 50 μL whole blood stained with antibodies in the TruCount tubes that contained a predefined number of fluorescent beads to provide internal calibration. The CD4[+] and CD8[+] T cell counts were then calculated using the ratio of CD4[+] and CD8[+] T cells to CD45[+] cells in the whole blood at the same time point.

Immunophenotyping of the immune cells isolated from blood, LNs, and intestine was performed using fluorescently conjugated mAbs (S3 and S4 Tables), chosen to characterize a wide range of immune cell types and markers of activation, proliferation, apoptosis and cellular homing. Live/Dead stains were included only for the innate cell panels (NKs, Mφs, and DCs). For the adaptive cell type panels, dead cells were omitted by gating first for singlets (S7 Fig). Data were acquired on an LSRII flow cytometer (Becton Dickinson, Franklin Lakes, NJ) and analyzed using the Flowjo software version 10.1r5 (Tree Star Inc, Ashland, OR). An example of the gating strategy used for delineating the major CD4[+] and CD8[+] T cell populations and subsets is shown in S7 Fig. In this flow analysis we did not include the colon samples, because the number of cells obtained from colon was inconsistent and did not always yield enough viable immune cells for staining.

## Testing the levels of plasma inflammatory cytokines and chemokines

Systemic changes in chemokines and cytokines were measured using an Invitrogen Monkey Cytokine Magnetic 29-Plex Panel (Invitrogen, Carlsbad, CA), as per the manufacturer's instructions using a Bio-Rad Bio-Plex Pro II Wash Station (Bio-Rad, Hercules, CA) and the Bio-Rad Bio-Plex 200 System (Bio-Rad) [69,74]. Plasma samples taken from both pre- and postinfection were assayed and used to calculate the total fold change in the plasma chemokine and cytokine levels. These fold change values were then normalized using a $log_2$ transformation. The transformed data was used to generate a heatmap via the publicly available Morpheus software from the Broad Institute (https://software.broadinstitute.org/morpheus/)

## Measuring plasma levels of markers of fibrosis, gut epithelial damage and microbial translocation

Systemic levels several markers of gut dysfunction were measured by ELISA, including: (i) LPS, which was established in the earliest study of HIV and microbial translocation as a means to measure microbial translocation levels (*Limulus* amebocyte assay, Cambrex, Rutherford, NJ) [51]; (ii) CRP, which is a well-established marker of inflammation (monkey CRP ELISA; Life Diagnostics, West Chester, PA) ([54,67]; sP-selectin, which serves both in thrombosis as well as recruitment of inflammatory leukocytes to sites of injury (Platinum ELISA; eBioscience, San Diego, CA) [79]; sCD14, which provides an alternate means to measure LPS levels and is linked to microbial translocation (Quantikine Human sCD14 Immunoassay; R&D Systems) [51,69,73]. All assays were performed as described previously [23,69].

Additionally, systemic levels of intestinal fatty acid binding protein (I-FABP) and hyaluronic acid (HA) were measured by conventional ELISA. I-FABP, a known marker of intestinal damage [151], was assessed using a Monkey I-FABP ELISA kit (MyBioSource, San Diego, CA), as described [144]. HA, a prognostic marker of fibrosis [99], was tested using a Monkey Hyaluronic Acid ELISA Kit (MyBioSource).

## Immunohistochemical assessment of mucosal tissues and quantitative image analysis

We examined the overall integrity of the gut mucosal barrier with a multifaceted set of markers to assess inflammation in the gut (MX1), epithelial cell proliferation (Ki-67), apoptosis (caspase-3), epithelium continuity (claudin-3) and microbial translocation (LPS-core and *Escherichia coli*). IHC and quantitative image analysis were performed as described [53,62,146]. Briefly, paraffin fixed tissues mounted on glass slides were deparaffinized by a battery of 3 x 5 min washes in xylene, then rehydrated by 3 x 5 min washes in 100%, 95% and 75% ethanol. Next, the tissues were boiled in diluted Antigen Unmasking buffer (Vector Laboratories, Burlingame, CA) for 25 min, allowed to cool to RT, then rinsed 5 min in 1xPBS, and incubated with 3% $H_2O_2$ for 15 min. After 2x5 min washes in 1xPBS, the tissues were blocked with Protein Block (Dako, Santa Clara, CA) for 30 min at RT, incubated with diluted primary antibody (S5 Table) (Dako Antibody Diluent) for at least 1 hr at RT, rinsed in 1xPBS for 2x5 min washes and incubated with diluted Vectastain secondary antibody (Vector Laboratories) for another 30 min at RT. After 2x5 min washes in 1xPBS, the tissues were incubated for 30 min at RT with Vectastain ABC solution (Vector Laboratories), followed by 2x7 min washes in 1xPBS, and addition of DAB diluted in DAB substrate (Dako) to generate a positive signal, at which point the reaction was quenched in deionized water for 5 min before being counterstained with haematoxylin. The tissues were then dehydrated with a reverse battery of ethanol and xylene (75%, 95% and 100%) before being coverslipped.

The tissues taken from the RMs were processed for IHC and stained in a similar fashion to the AGMs, as previously described [53,62]. To quantify the RM images, all stained tissue sections (n≥2 per animal) were scanned at high magnification (200x) using the Aperio AT2 system (Leica Biosystems, Wetzlar, Germany) yielding high-resolution images for the entire tissue section. Representative high magnification (200x) images (n≥20; 500 $\mu m^2$) were acquired from whole tissue scans and quantified.

For each of the AGMs, images of stained tissues were manually captured from stained tissues sections (n = 1 per tissue) using a computer-linked microscope camera (Zeiss, Oberkochen, Germany) and Axiovision software v.4.7 (Zeiss) at high magnification (200X). Sections were imaged for quantification from all AGMs included in this study. For quantification of immune activation, inflammation and apoptosis markers, images were quantified by measuring the % area of positive signal (n = 10–15 images; 5.15x10$^5$ $\mu m^2$). The quantified regions were randomly selected to minimize bias. The methods for total DAB quantification are given in S8 Fig and the methods for total collagen (Trichrome stained) quantification are given in S9 Fig. To quantify enterocyte proliferation, we measured the proportion of the villi length with Ki-67$^+$ epithelial cells, as described [53] (S10 Fig), by manually drawing a black line through the middle of the villi from the base to the luminal end, then bifurcating the line based on the position of the Ki-67$^+$ epithelial cells and measuring the two segments with the FIJI Wand tool. This allowed us to quantify the fraction of proliferating and activated mucosal epithelial cells, with 7–17 crypts being measured and averaged per AGM. To quantify the proliferation of cells specific to the lamina propria, we used FIJI to manually exclude the epithelium from the images (n = 4) from each AGM, allowing us to quantify the total % area of the positive signal in the lamina propria alone (S11 Fig). To quantify the percent of damaged epithelial barrier of the GI tract, we took multiple overlapping images (100X) of the gut epithelium stained for claudin-3 to create a large composite image (n>4 images per composite) of the GI tract using the Image Stitching plugin [152] in the FIJI software [153,154]. Once the composite image was generated, we first traced the length of the intact epithelium with a green line, and the damaged regions lacking claudin-3 staining with a red line. The total lengths of the two lines were then measured using the FIJI Wand tool (S12 Fig) to establish the length of intact *versus* broken epithelium, as previously described [62]. A total of 3–4 composite images were quantified for each AGM in this fashion. Similar methodologies were used for quantification of the IHC stains for the RMs, as previously described [62]. All AGM quantifications were done using the open-source FIJI software, v.1.0 [153,154].

## Assessment of the collagen deposition in the mucosal tissues

We assessed collagen deposition in the tissues using the Chromaview Advanced Testing Masson Trichrome Stain (Thermofisher) (S9 Fig), as per the manufacturer's instructions, with some modifications. Briefly, tissue slides were deparaffinized as done for IHC, and incubated overnight in Bouin's fluid. The slides were then washed repeatedly in deionized water, immersed in Working Weigert's Iron Haematoxylin Stain for 10 min, and rinsed in water for 5–10 min. The slides were transferred into Bierich's Scarlet-Acid Fuchsin Solution for 5 min before being rinsed in water for 30 sec and incubated 5 min with Phosphotungstic-Phosphomolybdic Acid solution. Finally, the slides were fully immersed in Aniline Blue Stain Solution for 30 min, washed 2x2 min with 1% acetic acid, and rinsed for 30 sec in deionized water. The slides were then dehydrated with 2x1 min washes in 100% ethanol, followed by 3x1 min washes in xylene before coverslipping. The collagen staining for the RMs was done in a similar fashion to the AGMs, as described [134].

## RNAseq transcriptomics

RNAseq and the subsequent bioinformatic analysis was performed as described [86]. Briefly, rectal tissues were immediately perfused in RNAlater and stored at -80˚C until further processing. Whole transcriptome libraries were constructed using the TruSeqStranded Total RNA with Ribo-Zero Gold (Illumina, San Diego, CA) as per the manufacturer's instructions. Libraries were quality controlled and quantitated using the BioAnalzyer 2100 system and qPCR (Kapa Biosystems, Woburn, MA). The resulting libraries were then sequenced initially on a HiSeq 2000 using HiSeq v3 sequencing reagents, with read number finishing using a Genome Analyzer IIx using GA v5 sequencing reagents, both of which generated paired end reads of 100 nucleotides (nt). The libraries were clonally amplified on a cluster generation station using Illumina HiSeq version 3 and GA version 4 cluster generation reagents to achieve a target density of approximately 700,000 (700K)/mm$^2$ in a single channel of a flow cell. Image analysis, base calling, and error estimation were performed using Illumina Analysis Pipeline (version 2.8).

## Statistical analysis

To compare changes in the immune cell populations between preinfection and postinfection time points from the same AGMs, we used the nonparametric two-tailed Wilcoxon matched-pairs signed ranks test, with a $p \leq 0.05$. For comparisons between control and infected AGMs, which included both flow cytometry results and the IHC quantifications, we used the unpaired nonparametric Kruskal-Wallis test, followed by a Dunn's multiple means comparison test to correct for multiple comparisons. A Mann-Whitney $U$ test was used to test differences between the set-point AGMs and the chronically infected RMs. The family-wise significance and confidence levels were set at 0.05. All tests were performed using the Graphpad Prism 6 Software (Graphpad Software Inc, La Jolla, CA).

## Supporting information

**S1 Table. Reference data for African green monkeys.**
(XLSX)

**S2 Table. Reference data for chronically SIV-infected rhesus macaques.**
(XLSX)

**S3 Table. Flow cytometry antibody panels.**
(XLSX)

**S4 Table. Flow cytometry antibodies.**
(XLSX)

**S5 Table. Primary antibodies for immunohistochemistry.**
(XLSX)

**S1 Fig. Animal groups for study of the early events of intrarectal SIVsab infection.** AGMs were serially euthanized throughout the acute and early chronic SIV infection and were divided into the following groups: (i) preinfection (baseline); (ii) preramp-up (1–3 dpi); (iii) ramp-up (4–6 dpi); (iv) peak (9–12 dpi); (v) set-point (46–55 dpi). Each group is assigned a corresponding color: orange (baseline), green (preramp), blue (ramp-up), red (peak) and yellow (set-point).
(TIF)

**S2 Fig. CD8$^+$ T-cell and CD20$^+$ populations in blood, jejunum and axillary lymph nodes (LNs) of SIVsab-infected AGMs.** Total populations of (A) CD8$^+$ T cells; and (B) CD20$^+$ B cells isolated from blood, axillary LN and jejunum. The values for blood represent absolute counts, while the values in the jejunum and LN represent percent populations. The five groups are based on the days postinfection, with: BL (baseline, preinfection), PRU (preramp, 1–3 dpi) RU (ramp-up, 4–6 dpi), PEAK (peak, 9-12dpi) and SP (set-point, 46–55 dpi). Each group is assigned a corresponding color: orange (baseline), green (preramp), blue (ramp-up), red (peak) and yellow (set-point). Asterisks indicates statistical significance when compared to baseline values, with $^* = p<0.05$; $^{**} = p<0.01$.
(TIF)

**S3 Fig. Innate immune cell populations in blood, jejunum and axillary lymph nodes (LNs) in SIVsab-infected AGMs.** The flow cytometry analysis encompassed multiple immune cell subtypes, including: (A) myeloid dendritic cells; (B) plasmacytoid dendritic cells; (C) monocytes/macrophages (CD14$^+$ CD163$^+$); and (D) natural killer cells (NKG2A$^+$). These cells were isolated from a variety of different tissues, including blood, jejunum and axillary LN. The five different time groups are based on the days postinfection, with: BL (baseline, preinfection), PRU (preramp, 1–3 dpi) RU (ramp-up, 4–6 dpi), PEAK (peak, 9-12dpi) and SP (set-point, 46–55 dpi). Each time group is assigned a corresponding color: orange (baseline), green (preramp), blue (ramp-up), red (peak) and yellow (set-point). Asterisks indicates statistical significance when compared to baseline values, with $^* = p<0.05$.
(TIF)

**S4 Fig. Plasma levels of markers of gut dysfunction and microbial translocation.** Plasma from each animal was tested using ELISA for: (A) lipopolysaccharide (LPS); (B) C-reactive protein (CRP); (C) soluble P-selectin (sP-selectin); and (D) soluble CD14 (sCD14). The values shown represent a total fold change in from baseline levels for each animal. The five groups are based on the days postinfection, with: BL (baseline, preinfection, orange), PRU (preramp, 1–3 dpi, green) RU (ramp-up, 4–6 dpi, blue), PEAK (peak, 9-12dpi, red) and SP (set-point, 46–55 dpi, yellow). Asterisks indicates statistical significance when compared to baseline values, with $^* = p<0.05$; $^{**} = p<0.01$.
(TIF)

**S5 Fig. Immunohistochemistries (IHC) of the jejunum in SIVsab-infected AGMs.** DAB-based IHC for the same array of markers that were used for the colon and axillary LN. In all the images, positive DAB signal is shown in brown, with the remaining tissue counterstained blue. Below are shown quantifications of positive signal within the image. The quantification for each animal represents the average of the values from 9–12 individual image quantifications. Villi enterocytes were included in all quantifications. The four different time groups are based on the days postinfection, with: BL (baseline, preinfection, orange), PRU (preramp, 1–3 dpi, green) RU (ramp-up, 4–6 dpi, blue), PEAK (peak, 9-12dpi, red) and SP (set-point, 46–55 dpi, yellow). All quantifications were performed using FIJI version—1.0. Asterisks indicate statistical significance p<0.05. All images were captured at 200X magnification using an AxioImager M1 bright-field microscope equipped with an AxioCam MRc5. Scale bar: 100 μm.
(TIF)

**S6 Fig. Gene expression in the gut over the course of SIV infection in AGMs and RMs.** RNAseq data from AGM and RM gut tissue displayed as a heatmap showing gene expression changes in specific genes of interest associated with biological processes related to SIV-infection and the host immune response. The level of alteration of gene expression is shown in blue (downregulation) and red (upregulation), with genes clustered using a Spearman correlation,

with the dendrogram showing relationship dispalyed on the left. Animal numbers are shown below the heatmap along with dpi, while the time groups are shown above the heatmap, with the colors indicating the groups: orange (, BL, baseline), green (PRU, preramp), blue (RU, ramp-up), red (peak), yellow (SP, set-point). The data for the time groups of the AGMs are listed in black text on the left, while the data from the equivalent RM time groups are listed in white text on the right.
(TIF)

**S7 Fig. Flow cytometry gating strategy for CD4$^+$ and CD8$^+$ populations and immune activation.** (A) Gating strategy used to delineate primary T-cell populations (CD3$^+$, CD4$^+$, CD8$^+$) T-cell populations. (B) Gating strategies to delineate the secondary T cell populations (EM, CM and naïve), Ki-67$^+$ T cells, CD69$^+$ T cells, and HLA-DR$^+$ CD38$^+$ T cells. All plots shown in (B) are CD4$^+$ T cells, but the same gating strategies were used for CD8$^+$ T cells. All gates were generated using Flowjo software version 10.1r5 (Tree Star Inc, Ashland, OR).
(TIF)

**S8 Fig. Method for quantification of positive DAB signal based on color deconvolution.** To quantify the DAB stain, each raw image (A) was processed using the Color Deconvolution 1.7 plugin for FIJI v.1.0. The preset DAB settings were selected, and the software separated the image into 3 color channels, with the brown channel representing the positive DAB signal (B). A threshold was then manually applied to the brown channel image to remove background coloration (C). Finally, the area of each image representing the DAB signal above threshold was measured as a percentage of the total area of the image. All images were captured at 200X magnification using an AxioImager M1 brightfield microscope equipped with an AxioCam MRc5. All image manipulations and measurements were done with FIJI v.1.0.
(TIF)

**S9 Fig. Method for quantification of tissue collagen based on color thresholds.** To quantify the amount of collagen in each tissue, each raw image (A) was processed with the built-in Color Threshold function in FIJI v.1.0. Using this feature, first the collagen was isolated from the rest of the image by adjusting the Hue value of the Color Threshold function to only encompass the blue of the collagen (B). It should be noted that the blue dye was also partially taken up by the goblet cells in the mucosal epithelium. After setting the Color Threshold, all background colors were removed, and the image was transformed into black and white (C). This eliminates all area of the image that is not blue coloration, which then can be measured by setting an intensity threshold to select all black area in the image. All images were captured at 200X magnification using an AxioImager M1 brightfield microscope equipped with an AxioCam MRc5. All image manipulations and measurements were done with FIJI v.1.0.
(TIF)

**S10 Fig. Approximation of epithelial proliferation in colonic crypts.** To generate an approximation of the levels of on-going proliferation in the gut epithelium, we captured images of individual crypts in tissue sections stained with Ki-67. As epithelial proliferation is localized to the base of the crypts, we measured both the total length of the crypt (total black line) and the length of the crypt with Ki-67$^+$ epithelial cells (black line below the bisecting green arrow). Since increased levels of proliferation should result in an increase in the total number of Ki-67$^+$ cells along the crypt, taking the ratio of the areas of the line allowed us to normalize proliferation across multiple crypts. All individual images were captured at 200X magnification using an AxioImager M1 brightfield microscope equipped with an AxioCam MRc5. All image manipulations and measurements were done with FIJI v.1.0.
(TIF)

**S11 Fig. Exclusion of epithelium to isolate cells in the lamina propria.** To measure the Ki-67 expression by cells in the lamina propria, all the epithelial cells were manually removed from the images. By overlaying white coloration on the epithelial sections of the crypts, they were excluded from the thresholding for positive DAB signal (S8 Fig). Then, a threshold could be applied to the area within the lamina propria alone and the total DAB signal measured as normal. All individual images were captured at 200X magnification using an AxioImager M1 brightfield microscope equipped with an AxioCam MRc5. All image manipulations and measurements were done with FIJI v.1.0.
(TIF)

**S12 Fig. Measurement of intact versus damaged transverse colon epithelium.** To determine the relative proportion of intact colon mucosal epithelium compared to damaged epithelium, multiple contiguous images of a section of colonic mucosa were obtained. The length of the epithelium shown in the generated composite image was then traced using FIJI v.1.0; here, green lines represents intact, continuous epithelium, while red lines indicates broken, discontinuous epithelium. The lines were drawn freehand at a constant width of 10 pixels. Following tracing, the area of the line segments was measured with FIJI and these areas were used to establish a ratio representing of intact *versus* broken epithelium. All individual images were captured at 100X magnification using an AxioImager M1 brightfield microscope equipped with an AxioCam MRc5. After collection, the images were stitched together to form a composite using the Stitching plugin for FIJI version 1.0.
(TIF)

## Acknowledgments

We would like to thank Drs. Claire Deleage, and Charles Rinaldo for their assistance and helpful suggestions. We would also like to thank to Drs. Nancy Miller, John Warren and Allan Schultz of the NIH and Dr. Samuel A. Levine of the University of Pittsburgh, who made this study possible. We also thank the University of Pittsburgh Statistics Consulting Center for their help with verifying the proper statistical methods for this study. We also thank all of our lab members who helped to finalize this publication.

## Author Contributions

**Conceptualization:** Kevin D. Raehtz, Jacob D. Estes, Cristian Apetrei, Ivona Pandrea.

**Data curation:** Kevin D. Raehtz, Fredrik Barrenäs.

**Formal analysis:** Kevin D. Raehtz, Fredrik Barrenäs, Cuiling Xu, Dongzhu Ma, Benjamin B. Policicchio, Michael Gale, Jr., Brandon F. Keele, Jacob D. Estes, Cristian Apetrei, Ivona Pandrea.

**Funding acquisition:** Cristian Apetrei, Ivona Pandrea.

**Investigation:** Kevin D. Raehtz, Fredrik Barrenäs, Cuiling Xu, Kathleen Busman-Sahay, Audrey Valentine, Lynn Law, Dongzhu Ma, Benjamin B. Policicchio, Viskam Wijewardana, Egidio Brocca-Cofano, Anita Trichel, Michael Gale, Jr., Brandon F. Keele, Jacob D. Estes, Cristian Apetrei, Ivona Pandrea.

**Methodology:** Kevin D. Raehtz, Michael Gale, Jr., Jacob D. Estes, Cristian Apetrei.

**Project administration:** Cristian Apetrei, Ivona Pandrea.

**Supervision:** Jacob D. Estes, Cristian Apetrei, Ivona Pandrea.

**Validation:** Michael Gale, Jr., Brandon F. Keele, Jacob D. Estes, Cristian Apetrei, Ivona Pandrea.

**Writing – original draft:** Kevin D. Raehtz, Fredrik Barrenäs, Cristian Apetrei, Ivona Pandrea.

**Writing – review & editing:** Kevin D. Raehtz, Cuiling Xu, Kathleen Busman-Sahay, Audrey Valentine, Lynn Law, Dongzhu Ma, Benjamin B. Policicchio, Viskam Wijewardana, Egidio Brocca-Cofano, Anita Trichel, Michael Gale, Jr., Brandon F. Keele, Jacob D. Estes, Cristian Apetrei, Ivona Pandrea.

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
