## [Decision Letter · Decision Letter 0]

3 Oct 2019

Dear Dr. Apetrei,

Thank you very much for submitting your manuscript "African Green Monkeys Avoid SIV Disease Progression by Preventing Intestinal Dysfunction and Maintaining Mucosal Barrier Integrity" (PPATHOGENS-D-19-01603) for review by PLOS Pathogens. Your manuscript was fully evaluated at the editorial level and by independent peer reviewers. The reviewers appreciated the attention to an important problem, but raised some substantial concerns about the manuscript as it currently stands. These issues must be addressed before we would be willing to consider a revised version of your study. We cannot, of course, promise publication at that time.

We therefore ask you to modify the manuscript according to the review recommendations before we can consider your manuscript for acceptance. Your revisions should address the specific points made by each reviewer.

(1) A letter containing a detailed list of your responses to the review comments and a description of the changes you have made in the manuscript. Please note while forming your response, if your article is accepted, you may have the opportunity to make the peer review history publicly available. The record will include editor decision letters (with reviews) and your responses to reviewer comments. If eligible, we will contact you to opt in or out.

(2) Two versions of the manuscript: one with either highlights or tracked changes denoting where the text has been changed; the other a clean version (uploaded as the manuscript file).

Additionally, to enhance the reproducibility of your results, PLOS recommends that you deposit your laboratory protocols in protocols.io, where a protocol can be assigned its own identifier (DOI) such that it can be cited independently in the future. For instructions see http://journals.plos.org/plospathogens/s/submission-guidelines#loc-materials-and-methods

We hope to receive your revised manuscript within 60 days. If you anticipate any delay in its return, we ask that you let us know the expected resubmission date by replying to this email. Revised manuscripts received beyond 60 days may require evaluation and peer review similar to that applied to newly submitted manuscripts.

[LINK]

Sincerely,

David T. Evans

Associate Editor

PLOS Pathogens

Susan Ross

Section Editor

PLOS Pathogens

Kasturi Haldar

Editor-in-Chief

PLOS Pathogens

orcid.org/0000-0001-5065-158X

Grant McFadden

Editor-in-Chief

PLOS Pathogens

orcid.org/0000-0002-2556-3526

Reviewer's Responses to Questions

**Part I - Summary**

Reviewer #1: In this manuscript, “African Green Monkeys Avoid SIV Disease Progression by Preventing Intestinal Dysfunction and Maintaining Mucosal Barrier Integrity”, Raehtz, K. et al. sought to characterize the immunological outcome and mucosal integrity of African Green Monkeys (AGMs) during SIV infection. AGMs are natural hosts for SIV and lack disease progression. The authors demonstrated that throughout the acute course of SIV infection, AGMs maintain mucosal barrier integrity, thereby mitigating the effects of microbial translocation (MT) and the associated chronic inflammation observed in SIV-infected Rhesus macaques (RMs). This is an important study that describes mechanisms of natural hosts controlling inflammation that, if translatable to humans, could lead to future therapeutic approaches to alleviate disease progression during the acute and early chronic phase of HIV infection. The experiments were conducted well, with analysis of mucosal integrity and activation at several key points in viral pathogenesis. The IHC data for maintenance of tight junctions and lack of fibrosis is very convincing, especially when shown cross-sectionally with RMs in the chronic phase of infection.

Other differences, however, are less clear, due to the way the data was reported. The manuscript could be improved by addressing points described below.

Reviewer #2: The authors address the important question about the role of microbial translocation in HIV infection. Previous studies have already provided evidence, that natural hosts of SIV do not display a disruption of the intestinal barrier nor microbial translocation. However, the previous studies have mostly focused on the chronic phase of infection. To better understand the mechanisms underlying the preservation of the intestinal barrier, here the early phase of infection, which is known to be crucial for the outcome of an infection, was analyzed. The authors performed an exhaustive study on many time points of early infection and in distinct body and intestinal compartments. The study includes phenotypical, transcriptomic and immunohistochemical approaches for the analysis of the tissues. By analyzing in detail the very early phase of infection, they detected two waves of systemic inflammation, the first one very early (days 3-4 p.i.) and the second one coinciding with the viremia peak in the natural hosts. A transcriptomic profiling provided interesting gene candidates that need to be further studied. The discussion is long but very well written. While this study in many parts remain descriptive, it defines parameters of the very early phases of infection in lymph nodes and gut of a nonpathogenic SIV infection, and attracts attention to the gut damage which, as has been shown by several studies, are not totally restored by anti-retroviral therapy in people infected by HIV and thus remains a major issue.

Reviewer #3: In this manuscript by senior authors who have extensively published on SIV in African Green monkeys (AGMs), the authors have conducted an exhaustive analysis of very early events post SIV infection to investigate the early changes in gut mucosal integrity and determine whether breaks in mucosal integrity occur early on and then revert to normal or never occur. Serial necropsies were performed on a total of 33 AGMs at time-points 1-3 days post infection, 4-6 dpi, 9-12 dpi and 46-55 dpi to correspond to preramp up, ramp up, peak viremia and early chronic infection. Findings are largely negative. Of major concern is that previously reported findings in natural hosts such as peripheral blood and gut CD4 depletion, acute immune activation are not observed in this study. The authors acknowledge this in the discussion and attribute it to the wide baseline inter-animal variability possibly masking differences in a group comparison. The analysis throughout the manuscript is shown as grouped comparisons of all animals at baseline vs the animals sacrificed at each time-point. Paired comparisons of baseline vs post SIV infection time-point of each cohort is warranted for all the immune parameters analyzed but is not shown.

Other comments:

1. Plasma viral load data should be shown. Figure 1 title states “Blood, jejunum and transverse colon tissue viral loads…..” but shows jejunum, transverse colon and axillary lymph node data.

2. The choice of tissues for viral load estimation is not provided. The reason for selecting axillary lymph node and not showing data on draining lymph nodes or rectal / descending colon tissue after intrarectal inoculation is not evident.

3. It would be helpful if the same y-axes scale is used for all the plots in Figure 1.

4. Figure 4 shows persistent elevated cytokine levels at the early chronic infection time-point (Figure 4), another unexplained discrepancy from previous reports in this model.

5. Rhesus or pig-tailed macaque comparison at similar time-points would have been very useful

Figure S4- shows ramp up and peak in macaques but this data is confined to gut transcriptomics. Viral RNA data and immune parameter data etc should also be shown for this group. The methods do not provide any information on the macaque cohort.

**Part II – Major Issues: Key Experiments Required for Acceptance**

Reviewer #1: 1. Figure 4 depicts fold changes of cytokines from baseline following SIVsab infection. It is unclear what “baseline” refers to here, since what indicated as baseline timepoint in the figure (0 dpi) already has a fold change. Is “baseline” the first access, D -50 pre-infection? If that is the case, how the authors interpret the presence of significant changes in the levels of several cytokines between 0 dpi and the “baseline” timepoint?

2. Similar comment applies to Figure 9, showing a heatmap of gene expression changes in the gut during SIV infection. It is unclear what they are comparing the pre-ramp, ramp-up, peak, and SP gene expression to, as they are showing baseline gene expression variation as well. Again, it is unclear what “baseline” refers to here.

a. The authors should also add the gene expression for MX1 to the main manuscript heatmap, currently MX1 is only shown via IHC and in Fig S4 and gene expression data would strengthen their claim.

b. Based on the RNAseq data in Fig S4, there also seems to be a significant amount MX1 mRNA expression during infection that remains elevated at SP. This is different to what is seen via IHC where at SP, MX1 returns to baseline value. Can the authors comment on this discrepancy?

3. When describing the lack of MT in AGMs it would also be advantageous to show levels of plasma LPS, LBP, sCD14, and EndoCAb which was described in the first MT paper by Brenchley, J. et al. Nat Med 2006. In this study, the authors utilized IHC to look at the core region of LPS and an anti-E. coli antibody to measure enterobacteria species; although that is a very important analyses, plasma levels of the above mentioned soluble markers will contribute to fully address a lack of MT in AGMs.

Reviewer #2: The authors claim that there were no changes in the levels of many immune cells in tissues and also no changes in immune activation except for the interferon responses. However, only one time point was analyzed after the peak viremia, and only 4 animals were analyzed at this time point (and only 2 animals regarding the innate immune cells). For some markers, these four animals displayed high inter-individual variability. From these observations it is therefore impossible to make general statements, such as on a lack of immune activation in the gut during SIV infection in natural hosts. It is possible that by analyzing additional time points after the peak and especially more animals at the time point after the peak, one would have detected transient increases in immune activation, such as of proliferation of T and B cells. General statements or extrapolations such as "little to no increase of additional markers of immune activation or inflammation occurred after infection" and "no significant alterations in genes related to immune activation...could be documented" should be removed from the abstract. This does not reduce the interest of the study and only increases its accuracy.

Reviewer #3: 1. Data re-analyzed to show paired comparisons rather than group analysis.

2. Comparison with macaques.

**Part III – Minor Issues: Editorial and Data Presentation Modifications**

Reviewer #1: 1. A schematic depicting collections and endpoints would be highly useful, as it avoids having the reader scrolling back to the intro to determine what the timepoints refer to in the subsequent figures.

2. Plasma viral loads were not shown in this study. It would be informative to show, particularly considering the variability in tissue viremia among the animals showed in Figure 1. For example, at peak of infection AGM24 was undetectable for both SIV-DNA and SIV-RNA in the transverse colon. Showing plasma viral loads would be helpful to compare infection between all animals.

3. Representative flow cytometry plots for Ki-67, CD69, and HLA-DR/CD38 expression should also be shown in the supplemental figures as this is one of the readouts for inflammation after infection.

4. Materials and methods:

a. Characteristics of the animals (RMs and AGMs) should be included in material and methods (sex, MHC status, age).

b. Line 716 is titled as “Tissue VLs quantification”, however, it was described as “Viral RNA was extracted from plasma…”.

c. List flow cytometry mAbs used for the study (fluorophore, clone, and company), similar to what the authors have for IHC antibodies (Table S3).

5. Figure 9. For the gene network pathway, downregulated genes should be shown in blue to complement the heatmap in Fig. 9A.

Reviewer #2: Line 621 : It is stated «that the lack of damage in the gut… are the main factors behind the control of immune activation .. and lack of disease progression in the natural hosts ». While the study confirms the lack of intestinal damage, this study cannot exclude that additional mechanisms are underlying the lack of disease. Several other, non-mutually exclusive potential mechanisms of protection in natural hosts have been described, and which are also mentioned by the authors in the introduction, such as the relative protection of Tcm, a mutation in the TLR4 gene, the control of viral replication in lymph nodes, the lack of fibrosis in lymph nodes etc. While microbial translocation might indeed be a major factor in driving chronic inflammation, this study is not designed to compare the impact of the multiple factors to each other and more caution is required.

Minor comments

Lines 137-141 : additional factors have been described, such as the use of CXCR6 that might explain the relative protection of central memory CD4+T cells from SIV infection in natural hosts (Wetzel et al).

Line 544 contains a citation of a manuscript that is in revision.

Indicate the scales on the images. Indicate how many animals and how many sections per tissue were studied for the image analyses.

Reviewer #3: (No Response)

PLOS authors have the option to publish the peer review history of their article (what does this mean?). If published, this will include your full peer review and any attached files.

Reviewer #1: No

Reviewer #2: No

Reviewer #3: No

---

## [Editor Report · Decision Letter 1]

18 Jan 2020

Dear Dr. Apetrei,

We are pleased to inform you that your manuscript 'African Green Monkeys Avoid SIV Disease Progression by Preventing Intestinal Dysfunction and Maintaining Mucosal Barrier Integrity' has been provisionally accepted for publication in PLOS Pathogens.

Before your manuscript can be formally accepted you will need to complete some formatting changes, which you will receive in a follow up email. A member of our team will be in touch within two working days with a set of requests.

Best regards,

David T. Evans

Associate Editor

PLOS Pathogens

Susan Ross

Section Editor

PLOS Pathogens

Kasturi Haldar

Editor-in-Chief

PLOS Pathogens

orcid.org/0000-0001-5065-158X

Michael Malim

Editor-in-Chief

PLOS Pathogens

orcid.org/0000-0002-7699-2064
---

## [Editor Report · Acceptance letter]

25 Feb 2020

Dear Dr. Apetrei,

We are delighted to inform you that your manuscript, "African green monkeys avoid SIV disease progression by preventing intestinal dysfunction and maintaining mucosal barrier integrity," has been formally accepted for publication in PLOS Pathogens.

Best regards,

Kasturi Haldar

Editor-in-Chief

PLOS Pathogens

orcid.org/0000-0001-5065-158X

Michael Malim

Editor-in-Chief

PLOS Pathogens

orcid.org/0000-0002-7699-2064